# Sprayable Hybrid Gel with Cannabidiol, Hyaluronic Acid, and Colloidal Silver: A Multifunctional Approach for Skin Lesion Therapy

**DOI:** 10.3390/pharmaceutics17091189

**Published:** 2025-09-12

**Authors:** Geta-Simona Cîrloiu (Boboc), Adina-Elena Segneanu, Ludovic Everard Bejenaru, Marius Ciprian Văruţ, Roxana Maria Bălăşoiu, Daniela Călina, Andreea-Cristina Stoian, Georgiana Băluşescu, Dumitru-Daniel Herea, Maria Viorica Ciocîlteu, Andrei Biţă, George Dan Mogoşanu, Cornelia Bejenaru

**Affiliations:** 1Doctoral School, University of Medicine and Pharmacy of Craiova, 2 Petru Rareş Street, 200349 Craiova, Romania; symonacirloiu@yahoo.ro; 2Institute for Advanced Environmental Research, West University of Timişoara (ICAM–WUT), 4 Oituz Street, 300086 Timişoara, Romania; adina.segneanu@e-uvt.ro; 3Department of Pharmacognosy & Phytotherapy, Faculty of Pharmacy, University of Medicine and Pharmacy of Craiova, 2 Petru Rareş Street, 200349 Craiova, Romania; andrei.bita@umfcv.ro (A.B.); george.mogosanu@umfcv.ro (G.D.M.); 4Department of Physics, Faculty of Pharmacy, University of Medicine and Pharmacy of Craiova, 2 Petru Rareş Street, 200349 Craiova, Romania; marius.varut@umfcv.ro; 5Department of Biochemistry, Faculty of Pharmacy, University of Medicine and Pharmacy of Craiova, 2 Petru Rareş Street, 200349 Craiova, Romania; roxana.balasoiu@umfcv.ro; 6Department of Clinical Pharmacy, Faculty of Pharmacy, University of Medicine and Pharmacy of Craiova, 2 Petru Rareş Street, 200349 Craiova, Romania; daniela.calina@umfcv.ro; 7Department of Infectious Diseases, Faculty of Medicine, University of Medicine and Pharmacy of Craiova, 2 Petru Rareş Street, 200349 Craiova, Romania; andreea.stoian@umfcv.ro; 8National Institute of Research and Development for Technical Physics, 47 Dimitrie Mangeron Avenue, 700050 Iaşi, Romania; georgiana.balusescu@tuiasi.ro (G.B.); dherea@phys-iasi.ro (D.-D.H.); 9Department of Instrumental and Analytical Chemistry, Faculty of Pharmacy, University of Medicine and Pharmacy of Craiova, 2 Petru Rareş Street, 200349 Craiova, Romania; maria.ciocilteu@umfcv.ro; 10Department of Pharmaceutical Botany, Faculty of Pharmacy, University of Medicine and Pharmacy of Craiova, 2 Petru Rareş Street, 200349 Craiova, Romania; cornelia.bejenaru@umfcv.ro

**Keywords:** hyaluronic acid, cannabidiol, colloidal silver, thermoresponsive hybrid gel, antimicrobial efficacy, biocompatibility, wound healing

## Abstract

**Background/Objectives**: This study presents the development and characterization of a novel thermoresponsive hydrogel composed of hyaluronic acid (HA), poloxamer 407, cannabidiol (CBD), and colloidal silver (Ag), designed for topical antimicrobial therapy. **Methods**: The Ag-CBD complex was first synthesized and subsequently incorporated into a HA–poloxamer gel matrix to produce a stable, sprayable formulation with suitable physicochemical properties for dermal applications. **Results**: The HA-Ag-CBD hybrid gel exhibited a physiological pH, a gelation temperature compatible with skin surface conditions, and favorable rheological behavior, including thixotropy and shear thinning—critical for uniform application and retention under dynamic conditions. Release studies confirmed a sustained delivery profile, supporting prolonged local activity of CBD and colloidal Ag. Antimicrobial assays demonstrated that the HA-Ag-CBD hybrid gel retained potent activity against *Staphylococcus aureus* and *Candida albicans*, with minimum inhibitory and bactericidal concentrations (MIC/MBC) statistically comparable to those of the unencapsulated Ag-CBD complex. Against *E. coli*, the HA-Ag-CBD hydrogel exhibited primarily bacteriostatic activity, with a low MIC (9.24 μg/mL) but a substantially higher MBC (387.35 μg/mL), consistent with the intrinsic structural resistance of Gram-negative bacteria. In contrast, bactericidal activity was more pronounced against Gram-positive strains, reflecting differential susceptibility related to bacterial envelope properties. CBD consistently demonstrated superior antimicrobial efficacy to colloidal Ag, while the Ag-CBD combination produced slightly enhanced, mainly additive effects, likely due to complementary membrane disruption and intracellular Ag^+^ ion activity. Cytotoxicity assays on normal human dermal fibroblasts confirmed that the HA-Ag-CBD hybrid gel maintained >70% cell viability at therapeutically relevant concentrations, in accordance with ISO 10993-5:2009 guidelines, and effectively mitigated the inherent cytotoxicity of the Ag-CBD complex. **Conclusions**: The HA-Ag-CBD hybrid gel demonstrates strong potential as a biocompatible, multifunctional topical formulation for the treatment of infected wounds and skin lesions. Future work will focus on in vivo evaluation, assessment of skin permeation, and further development to support translational applications.

## 1. Introduction

Skin lesions arising from trauma, infection, inflammation, or chronic pathological conditions represent a significant clinical challenge, necessitating therapeutic approaches that can simultaneously control microbial contamination, alleviate inflammation, and promote tissue regeneration. The multifaceted process of wound healing demands advanced topical systems designed not only to accelerate repair but also to reduce patient discomfort and prevent complications. Accordingly, modern dermatological strategies increasingly focus on multifunctional delivery platforms that incorporate bioactive agents with complementary mechanisms of action, enhancing therapeutic efficacy and patient adherence [1,2].

Gels provide versatile platforms with broad applications beyond biomedical use, offering ease of administration, controlled release, and enhanced bioavailability. Their capacity to incorporate diverse active agents makes them ideal for wound healing, infection control, and inflammation management, as well as cosmetic, agricultural, and environmental purposes such as skin care, plant protection, and pollutant remediation [3,4].

Topical gels have emerged as a versatile and advanced platform for localized drug delivery, offering several practical advantages, including ease of application, improved patient compliance, and the capacity to deliver both hydrophilic and lipophilic agents directly to the target site. Their high water content ensures effective skin hydration and a soothing cooling effect, while their semi-solid consistency facilitates prolonged skin contact and controlled release of therapeutic compounds. Thermoresponsive gels, in particular, offer significant clinical benefits by transitioning from a liquid state at ambient temperature to a gel at physiological skin temperature, thereby minimizing leakage and enhancing localized retention. These properties make gels ideally suited for the treatment of wounds, infections, burns, and inflammatory dermatoses, where maintaining a moist, protective, and bioactive environment is critical for optimal healing outcomes [5,6,7,8,9].

Among thermoresponsive materials, poloxamers (Pluronics) are synthetic triblock copolymers consisting of hydrophilic poly(ethylene oxide) (PEO) and hydrophobic poly(propylene oxide) (PPO) blocks arranged in a PEO–PPO–PEO architecture. Their thermoresponsive nature arises from reversible micelle formation and gelation in aqueous environments, strongly influenced by polymer concentration and temperature. Poloxamer 407 is notable for its ability to form gels in situ at near-physiological temperatures (30–37 °C), making it highly attractive for biomedical applications such as topical and injectable drug delivery systems (DDSs). In contrast, poloxamer 188 contains less hydrophobic PPO content, exhibiting diminished gelation properties. The biocompatibility, non-irritancy, and reversible sol–gel transitions of poloxamers underpin their widespread use in pharmaceutical and dermatological formulations where temperature-triggered release is desired [10,11,12].

Hyaluronic acid (HA) is a naturally occurring glycosaminoglycan abundant in the extracellular matrix, well-recognized for its exceptional water retention, viscoelasticity, and intrinsic biocompatibility. HA plays a pivotal role in skin physiology by maintaining tissue hydration, facilitating cell migration, and supporting epidermal regeneration. These properties make HA an invaluable component for topical gel formulations, as it enables mild hydrogel formation that encapsulates and sustains the release of sensitive bioactive molecules. Furthermore, HA’s inherent mucoadhesive and anti-inflammatory effects enhance its therapeutic potential in wound healing and in the management of burns, infections, and chronic skin disorders [13,14,15,16].

Cannabidiol (CBD), the second most abundant phytocannabinoid derived from *Cannabis sativa*, has garnered substantial interest as a bioactive agent with broad therapeutic potential. Unlike Δ^9^-*trans*-tetrahydrocannabinol (Δ^9^-THC), the primary psychoactive cannabinoid, CBD is non-psychoactive and engages a diverse array of molecular targets, including cannabinoid receptors, nuclear receptors, ion channels, and enzymes involved in inflammation and neuroprotection [17]. This diverse pharmacology underlies CBD’s demonstrated neuroprotective, analgesic, anti-inflammatory, antioxidant, anxiolytic, and antimicrobial effects. Clinically, purified CBD formulations such as EPIDIOLEX^®^ have gained regulatory approval for refractory epilepsy, and emerging evidence supports its expanding applications in dermatology and systemic diseases [17,18,19,20]. Despite increasing commercialization, stringent quality controls remain essential to ensure the safety and efficacy of CBD products [21].

Colloidal silver (Ag), composed of nanoscale Ag particles suspended in water, has been used historically for wound care and infection control. Modern research confirms its broad-spectrum antimicrobial activity against bacteria (including resistant strains), fungi, and some viruses, mediated primarily through the controlled release of Ag^+^ ions, which disrupt microbial membranes, inhibit enzymatic systems, and induce oxidative stress. In addition to antimicrobial effects, colloidal Ag exerts anti-inflammatory activity by modulating proinflammatory cytokines, thereby creating a regenerative environment conducive to healing. Its tolerability and low propensity for resistance development position colloidal Ag as a valuable topical agent in advanced wound management [22,23,24].

Leveraging the synergistic therapeutic potential of HA, CBD, and colloidal silver nanoparticles (AgNPs), this study introduces a novel thermosensitive gel formulation specifically engineered for the advanced topical treatment of skin lesions. Distinctively, the gel maintains a fluid, sprayable state at temperatures below 22 °C, enabling precise, non-invasive application over affected areas. Upon exposure to physiological skin temperatures, it undergoes an immediate and reversible sol–gel transition, forming a stable semi-solid matrix that securely adheres to the lesion site. This unique thermoresponsive behavior not only prevents runoff and enhances patient compliance but also facilitates sustained, localized release of multiple bioactive agents with complementary antimicrobial, anti-inflammatory, and regenerative functions.

Although HA-based hydrogels, CBD topical formulations, and AgNP-loaded biomaterials have each been independently explored, these approaches have largely remained single-function systems. HA hydrogels are valued for hydration and regenerative support but lack intrinsic antimicrobial or anti-inflammatory activity [25].

CBD formulations demonstrate anti-inflammatory and antioxidant effects yet provide minimal antimicrobial protection and often suffer from poor stability in aqueous carriers [17,26,27].

AgNP-containing hydrogels exhibit broad-spectrum antimicrobial action but fail to address oxidative stress and inflammation, which are equally critical in wound healing [28,29].

Moreover, while thermoresponsive poloxamer gels have been studied as versatile drug carriers, they have rarely been engineered to deliver multiple complementary bioactives [30,31].

Hybrid drug delivery systems, which integrate organic carriers such as biopolymers with inorganic or bioactive nanocomponents, have emerged as a promising solution to these limitations [32,33,34,35].

By combining complementary functionalities, such systems provide synergistic therapeutic effects, protect labile molecules like CBD from degradation, and allow tunable release kinetics while maintaining biocompatibility and thermoresponsiveness. This hybrid design paradigm therefore offers a rational strategy for creating multifunctional topical platforms that surpass the constraints of single-component formulations.

To the best of our knowledge, no prior study has combined HA, CBD, and AgNPs into a single thermoresponsive, sprayable hydrogel platform designed to simultaneously deliver antimicrobial, antioxidant/anti-inflammatory, and regenerative benefits with improved clinical practicality through temperature-triggered sol–gel transition. This integrated strategy establishes a novel multifunctional system with potential advantages over existing mono-component formulations.

## 2. Materials and Methods

### 2.1. Chemicals and Reagents

Poloxamer 407, poloxamer 188, HA, CBD, and colloidal Ag (density, 10.49 g/cm^3^) were purchased from Sigma-Aldrich (Taufkirchen, Germany). Purified water was purchased from Merck Millipore (Darmstadt, Germany).

### 2.2. Bacterial and Fungal Strains

The study utilized several bacterial strains sourced from the American Type Culture Collection (ATCC; Manassas, VA, USA), including *Staphylococcus aureus* (ATCC 29213), *Escherichia coli* (ATCC 25922), and *Pseudomonas aeruginosa* (ATCC 27853), as well as a fungal strain, *Candida albicans* (ATCC 14053).

### 2.3. Cell Lines

Primary normal human dermal fibroblasts (NHDFs) were obtained from the ATCC (Manassas, VA, USA). These cells were selected for their well-characterized phenotype and relevance as a model for dermal tissue studies. NHDFs were maintained under standard culture conditions at 37 °C with 5% carbon dioxide (CO_2_) in a humidified incubator to ensure optimal growth and viability.

The culture medium consisted of Dulbecco’s Modified Eagle’s Medium (DMEM; Gibco, Thermo Fisher Scientific, Leicestershire, UK) supplemented with 10% (v/v) heat-inactivated fetal bovine serum (FBS; Gibco, Thermo Fisher Scientific) to provide essential growth factors and nutrients. To prevent microbial contamination, the medium was further supplemented with 1% (v/v) antibiotic–antimycotic solution containing 100 U/mL of penicillin, 100 μg/mL of streptomycin, and 0.25 μg/mL of amphotericin B (Sigma-Aldrich, St. Louis, MO, USA). Cells were routinely passaged at 80–90% confluence using 0.25% trypsin–ethylenediaminetetraacetic acid (EDTA) (Gibco, Thermo Fisher Scientific) and seeded at a density of 5 × 10^3^ cells/cm^2^ for experimental assays. All procedures were performed under sterile conditions in a Class II biosafety cabinet to ensure the reproducibility and reliability of the results.

### 2.4. Formulation

The hydrogels were prepared under controlled low-temperature conditions to preserve the stability of thermolabile constituents and prevent nanoparticle (NP) aggregation. Cold purified water was pre-equilibrated and maintained at 4–8 °C in a refrigerated chamber. All subsequent steps, including HA incorporation and addition of the Ag-CBD dispersion, were carried out in an ice bath with continuous monitoring, ensuring that the temperature remained consistently below 15 °C throughout the process.

Poloxamer 407 and poloxamer 188 were accurately weighed and slowly dispersed in cold water under magnetic stirring to achieve complete solubilization while minimizing air entrapment. The solution was refrigerated overnight (≥12 h) to ensure full polymer hydration, yielding a clear, homogeneous base. HA was then gradually added under continuous slow stirring at low temperatures to prevent lump formation and ensure uniform distribution. Separately, CBD was dissolved in 4% (v/v) ethanol and gently mixed with colloidal AgNPs (0.02%) to form a stable Ag-CBD dispersion. This mixture was incorporated into the HA–poloxamer base at <15 °C, promoting even distribution, emulsification, and preservation of molecular integrity (Table 1; Figure 1a–c). The final hydrogel was transferred into opaque, airless containers to protect light-sensitive components and stored at 4–8 °C until use.

The CBD concentration (0.5%, w/v) was selected based on prior reports demonstrating therapeutic efficacy of topical CBD in the 0.1–1.0% range [17,18,19,20,21,26,27] while ensuring stability and compatibility with the HA–poloxamer–AgNP system. Preliminary optimization confirmed that 0.5% provided an optimal balance between antioxidant/anti-inflammatory activity and hydrogel integrity.

The incorporation of CBD and AgNPs into the HA–poloxamer hydrogel was confirmed through complementary characterization techniques: Fourier-transform infrared (FTIR) analysis to verify the presence of CBD functional groups within the gel matrix, scanning electron microscopy (SEM) imaging to confirm the homogeneous AgNP distribution, and dynamic light-scattering (DLS) measurements to display NP stability and preserved hydrodynamic size. These analyses were selected to support the successful loading and structural integrity of both actives, although quantitative release into biological media was not measured in this study.

All hydrogel batches were prepared following a controlled low-temperature protocol, with strict monitoring of temperature and mixing. Reproducibility was verified by assessing particle distribution and hydrogel morphology using SEM and DLS, ensuring consistent appearance and rheological behavior across independent replicates.

### 2.5. pH Measurement of Hydrogel

The pH of the final HA-Ag-CBD hydrogel was measured at room temperature (22–25 °C) using a calibrated WTW^®^ VARIO 2V00 digital pH meter (Xylem Analytics GmbH & Co. KG, Weilheim, Germany). Measurements were performed in triplicate on freshly prepared samples to ensure reproducibility. The hydrogel consistently exhibited a pH range of 6.8–7.4, confirming its compatibility with physiological and topical skin conditions [36,37].

### 2.6. Fourier-Transform Infrared Spectroscopy

FTIR spectra were acquired using a Shimadzu AIM-9000 spectrometer with attenuated total reflectance (ATR) accessories (Shimadzu Corp., Tokyo, Japan). Spectra were recorded over 20 scans at a resolution of 4 cm^−1^ in the range of 4000–400 cm^−1^. Wavelength assignments were based on the literature analysis.

### 2.7. Scanning Electron Microscopy Analysis

Surface morphology was investigated using a JSM-IT200 InTouchScope™ scanning electron microscope (JEOL Ltd., Freising, Germany) equipped with a field emission gun (FEG) and an energy-dispersive X-ray spectroscopy (EDS) system. Samples were mounted on carbon tape and sputter-coated with a 10 nm gold layer using a Quorum Q150R ES sputter coater (Quorum Technologies Ltd., Laughton, UK). The coating chamber was evacuated to ~0.1 mbar, and sputtering was performed at 40 mA for 60 s to achieve a uniform conductive layer while preserving nanoscale surface features, optimizing resolution and contrast. Imaging was carried out at an accelerating voltage of 15 kV and a working distance of 10 mm, with magnifications ranging from 100× to 10,000×. EDS analysis was performed to quantify elemental distributions, with spectra collected from multiple regions to ensure representativeness.

### 2.8. Dynamic Light-Scattering Analysis

Particle size distribution (PSD) of the hydrogel samples was measured using a Microtrac Nanotrac Wave II (Microtrac Retsch GmbH, Montgomeryville, PA, USA). Samples were dispersed in deionized water at a concentration of 0.1 mg/mL to minimize multiple scattering and ensure colloidal stability. Measurements were performed in batch mode at 23 °C, using a laser wavelength of 780 nm and a scattering angle of 180°, with each sample equilibrated for 2 min prior to acquisition. All measurements were conducted in triplicate, and results are reported as the mean hydrodynamic diameter and polydispersity index (PDI). This protocol ensures reproducible PSD and PDI across independent hydrogel batches.

### 2.9. Rheological Analysis

The rheology of the liquid was investigated by using a Thermo Haake 500 viscosimeter (Thermo Fisher Scientific Inc., Waltham, MA, USA) with a thermostat in order to obtain the rheograms for each temperature. Measurement of viscosity was performed, and after 5 min, the corresponding temperature was obtained. Throughout the time needed to wait for the temperature to settle, the rotational speed of the rotor was maintained at very low revolutions per minute (RPM) in order to maintain a uniform distribution of the liquid properties. After the required temperature was obtained and the waiting period expired, the rotational speed was increased, and after 5 s, the rheological parameters were obtained. The rotational speed increased from the first rotational speed (5 RPM) to the last (600 RPM). The shear rate (1/s) for each rotational step was obtained from the instrument and used to create rheograms.

### 2.10. In Vitro Release Profile of CBD and AgNPs

The in vitro release of CBD and AgNPs from the Ag-CBD complex and the HA-Ag-CBD hybrid hydrogel was investigated using a modified membraneless diffusion method [38,39].

For the hydrogel, 1.5 mL of the HA–poloxamer sol containing CBD and AgNPs was transferred into pre-weighed flat-bottomed vials, allowed to gel at 37 °C in a water bath, and overlaid with 2 mL of pre-heated phosphate-buffered saline (PBS; pH 7.4) containing 5% v/v ethanol to maintain sink conditions. For the Ag-CBD complex, 1.5 mL of colloidal dispersion (0.5% w/v CBD, 0.02% w/v AgNPs in 4% v/v ethanol) was placed in identical vials without gelation. All vials were incubated at 37 °C under shaking (100 ± 10 RPM).

At predetermined time points (0.5, 1, 2, 4, 8, 12, 24, and 48 h), aliquots (1.0 mL) of the release medium were withdrawn and replaced with an equal volume of fresh pre-warmed PBS/ethanol solution to maintain a constant volume and concentration gradient. The collected samples were centrifuged (10,000 RPM, 10 min, at 4 °C) to remove residual HA–poloxamer matrix fragments and analyzed by ultraviolet–visible (UV–Vis) spectrophotometry (CBD at 274 nm and AgNPs at 417 nm) using validated calibration curves prepared under identical solvent conditions (PBS containing 5% v/v ethanol) [40,41].

Hydrogel dissolution was quantified gravimetrically by weighing dried vials after complete release.

All experiments were conducted in triplicate (n = 3), and the results are presented as the mean ± standard deviation (SD). Cumulative release profiles were analyzed using the Korsmeyer–Peppas model to characterize the underlying release mechanism (Equation (1)):(1)$$\frac{M_{t}}{M_{\infty }}=kt^{n}$$
where $\frac{M_{t}}{M_{\infty }}$ is the fraction of the drug released at time *t* (h); *k* is the kinetic constant; and n is the release exponent, indicating the mechanism of drug transport.

### 2.11. Antimicrobial Activity

The antimicrobial efficacy of the HA-Ag-CBD hybrid gel was evaluated using standardized in vitro assays against three bacterial reference strains—*S. aureus* (ATCC 29213), *E. coli* (ATCC 25922), and *P. aeruginosa* (ATCC 27853)—and one fungal reference strain, *C. albicans* (ATCC 14053). Bacterial and fungal inocula were adjusted to a turbidity equivalent of the 0.5 McFarland standard and uniformly spread onto Müller–Hinton agar plates within 15 min of preparation. Sterile 6 mm paper disks (Thermo Fisher Scientific Inc., Oxford, UK) were impregnated with test samples and aseptically placed onto the agar surface using sterilized forceps. Sterile water served as the negative control. The plates were incubated at 37 °C for 24 h. Antimicrobial activity was quantified by measuring the inhibition zone (IZ) diameters in millimeters, including the 6 mm disk.

Minimum inhibitory concentration (MIC) and minimum bactericidal/fungicidal concentration (MBC/MFC) assays were conducted for CBD, colloidal Ag, the Ag-CBD complex, and the HA-Ag-CBD hydrogel. Stock solutions were prepared in the appropriate broth medium and serially diluted two-fold to achieve a concentration range of 1–400 μg/mL of active compound equivalents. The maximum tested concentration (400 μg/mL) ensured coverage above all inhibitory thresholds for the selected microbial strains. For disk diffusion assays, 10 μL of each solution was applied per disk, with the deposited amount of active compound calculated directly from the stock concentration. All reported concentrations for MIC, MBC/MFC, and disk diffusion refer exclusively to active compound equivalents rather than the total formulation mass. This strategy enables precise quantification of antimicrobial effects, ensures reproducibility, and facilitates direct comparison between individual agents and hybrid formulations.

To evaluate the potential intrinsic antimicrobial effects of the delivery vehicle, the HA–poloxamer gel matrix was included as a negative control in all assays. The blank gel was independently tested against Gram-positive, Gram-negative, and fungal strains using both disk diffusion and broth microdilution methods. These controls confirm that the antimicrobial effects observed in the HA-Ag-CBD formulation are attributable solely to the incorporated active agents and not to the gel matrix.

### 2.12. Cell Viability Assay

Assessing cytotoxicity on NHDFs is a critical step in evaluating the biocompatibility of compounds intended for topical delivery or skin-contact applications. NHDFs, as key mediators of skin homeostasis, collagen production, and tissue regeneration, provide a biologically relevant in vitro model for predicting cellular responses to candidate formulations.

Cytotoxicity was evaluated using the 3-(4,5-dimethylthiazol-2-yl)-2,5-diphenyltetrazolium bromide (MTT) assay, measuring cell viability across a concentration range of 25–100 μg/mL at three exposure durations (24, 48, and 72 h). This time- and dose-dependent analysis enables comprehensive assessment of how the hydrogel and its individual components influence fibroblast viability under conditions simulating both acute and prolonged skin exposure.

Half-maximal inhibitory concentration (IC_50_) values were calculated for each timepoint from triplicate measurements using nonlinear regression of the concentration–response curves, providing quantitative evaluation of cytotoxic potency. Although direct quantification of CBD and AgNP release from the hydrogel was not performed in this study, the selected concentration range corresponds to therapeutically relevant doses, based on prior in vitro antimicrobial efficacy and preliminary cytotoxicity screening.

These analyses allow direct comparison of biological safety profiles, identification of safe concentration thresholds, and evaluation of the potential formulation-dependent modulation of toxicity. Collectively, the cell viability and IC_50_ data establish a robust framework for assessing the in vitro biocompatibility of both free and hydrogel-encapsulated active agents.

### 2.13. Statistical Analysis

All experiments were performed in triplicate, with results reported as the mean ± SD. Statistical significance was determined using Student’s *t*-test for pairwise comparisons and one-way analysis of variance (ANOVA), followed by Tukey’s honestly significant difference (HSD) post hoc test for multiple pairwise comparisons, analyzed using Microsoft Excel 2019 version 16.31 (Microsoft Corporation, Redmond, WA, USA). A *p*-value < 0.05 was considered statistically significant.

## 3. Results

The HA-Ag-CBD hydrogel was first visually assessed to evaluate its sol-to-gel transition, an essential feature for topical applications. As shown in Figure 1a–c, the hydrogel exists as a clear, easily handled liquid prior to gelation (Figure 1a) and forms a stable, self-supporting gel at 37 °C (Figure 1b,c). This macroscopic behavior demonstrates the hydrogel’s suitability for controlled delivery and provides context for subsequent characterization. Following this, the hydrogel was systematically analyzed using FTIR, SEM, DLS, and rheology to elucidate its chemical structure, morphology, particle distribution, and mechanical properties.

### 3.1. Fourier-Transform Infrared Spectroscopy Analysis

FTIR spectroscopy was employed to rigorously assess the chemical integrity, molecular interactions, and structural stability of the newly prepared hydrogel system, which incorporates CBD and colloidal Ag within a HA gel (HA–poloxamer) matrix (Figure 2a). The spectral data presented in Figure 2b,c provide detailed evidence of characteristic vibrational bands, revealing shifts and interactions that confirm the successful integration of each component.

The FTIR spectrum of colloidal Ag (Figure 2b) displays a broad O–H stretching band at 3440 cm^−1^, attributed to surface hydroxyl (–OH) groups or adsorbed water, typical of colloidal systems. Aliphatic C–H stretching vibrations appear at 2926 cm^−1^ and 2855 cm^−1^, while distinct asymmetric and symmetric carboxylate (–COO^−^) stretching bands at 1635 cm^−1^ and 1383 cm^−1^ suggest coordination of an organic stabilizing agent at the Ag surface. Additional peaks at 1111 cm^−1^ (C–O stretching) and 618 cm^−1^ (metal–oxygen interactions) further support the presence of functional groups contributing to the colloidal Ag’s surface chemistry and stability, in agreement with the literature data [42].

The FTIR spectrum of the Ag-CBD complex (Figure 2b) reveals notable spectral shifts and the emergence of characteristic bands, confirming the successful incorporation of CBD onto the colloidal Ag surface. The broad O–H stretching vibration shifts slightly to 3450 cm^−1^, indicative of enhanced hydrogen bonding interactions, possibly between CBD’s –OH groups and the Ag surface. A weak aromatic C–H stretching band appears at 3006 cm^−1^, while the aliphatic C–H asymmetric and symmetric stretching bands intensify at 2916 cm^−1^ and 2855 cm^−1^, respectively, corresponding to the hydrocarbon backbone of CBD. The carboxylate stretching vibrations also undergo a bathochromic shift, appearing at 1646 cm^−1^ and 1424 cm^−1^, which suggests specific coordination of CBD’s phenolic or carboxylic moieties with the Ag surface. The emergence of a new peak at 1323 cm^−1^ is assigned to C–C aromatic stretching of the phenolic ring, further indicating CBD’s structural integrity within the complex. Additional fingerprint bands observed at 1162 cm^−1^ and 1021 cm^−1^ (C–O stretching), along with peaks at 953 cm^−1^, 668 cm^−1^, and 472 cm^−1^, are consistent with the vibrational profile of CBD. Collectively, these spectral modifications, shifts, intensity changes, and new band appearances provide compelling evidence for the successful binding of CBD to the colloidal Ag surface while preserving its molecular framework.

The FTIR spectrum of the final HA-Ag-CBD hybrid gel (Figure 2c) displays the combined spectral features of all constituents, confirming their successful and cohesive integration within the hydrogel matrix. A broad O–H stretching band centered at 3433 cm^−1^ persists, reflecting the presence of –OH groups and extensive hydrogen bonding within the hydrated polymeric network. Aliphatic C–H stretching bands appear at 2979 cm^−1^, 2929 cm^−1^, and 2868 cm^−1^, characteristic of the polymer backbone and the CBD hydrocarbon moieties. The complete disappearance of the free HA carboxylic acid peak at 1748 cm^−1^, along with the appearance of coordinated carboxylate bands at 1646 cm^−1^ and 1457 cm^−1^, strongly suggests the involvement of electrostatic interactions or hydrogen bonding between the carboxyl groups of HA and functional moieties of both colloidal Ag and CBD. The overlapping peaks at 1356 cm^−1^, 1296 cm^−1^, and 1255 cm^−1^ are attributed to the combined contributions of the HA gel (HA–poloxamer) matrix and CBD, further supporting their structural association. Preservation of distinct fingerprint bands at 1104 cm^−1^, 1024 cm^−1^, 953 cm^−1^, 711 cm^−1^, and 476 cm^−1^ confirms the chemical integrity of both CBD and HA within the new hybrid gel. Collectively, these spectral features provide robust evidence for the stable and homogeneous incorporation of colloidal Ag and CBD into the HA-based hydrogel matrix, forming a multifunctional network with retained molecular characteristics of all components [43,44,45].

### 3.2. Scanning Electron Microscopy Analysis

SEM was employed to evaluate the surface morphology and nanoscale distribution of components within the new HA-Ag-CBD hybrid gel. This technique provides high-resolution imaging critical for confirming the successful incorporation of the Ag-CBD complex, assessing particle dispersion, and characterizing the structural integrity of the hydrogel matrix.

Figure 3a–d presents SEM micrographs highlighting the morphological profiles of CBD, Ag-CBD, the HA gel (HA–poloxamer) matrix, and the HA-Ag-CBD hybrid gel. These high-resolution images provide clear insight into the distinct micro- and nanoscale features of each component, revealing critical structural differences that reflect their individual compositions and interactions within the composite system.

SEM imaging of the CBD sample (Figure 3a) reveals an irregular, semi-amorphous surface morphology with heterogeneous structures at the nanoscale. The observed features likely reflect dried lipidic residues and dispersed organic phases characteristic of oil-based formulations, consistent with the encapsulated nature of the CBD product.

The micrograph of the Ag-CBD sample (Figure 3b) reveals a core–shell-like morphology, with a dark central region corresponding to AgNPs (50–150 nm), surrounded by bright, irregular, and fibrous surface structures attributed to the CBD sample.

The uniform particle distribution and well-defined architecture suggest controlled assembly and effective surface interaction.

This contrast indicates the successful association of CBD with the colloidal Ag surface. The relatively uniform particle distribution and defined architecture suggest effective surface interaction and stabilization. The coexistence of crystalline features within the CBD layer and semi-amorphous interfacial regions indicates strong interfacial interactions between the organic CBD layer and the inorganic Ag core, which enhance the stability of the new Ag-CBD complex. Overall, the SEM data support the formation of a nanoscale Ag-CBD complex with structural coherence.

Conversely, SEM analysis of the HA hydrogel (Figure 3c) reveals a porous nanoscale architecture composed of irregular semi-crystalline clusters and amorphous domains, corresponding to the HA and poloxamer regions, respectively. This well-organized network, stabilized by poloxamer micelles, exhibits particle sizes ranging from tens to hundreds of nanometers, indicating uniform dispersion and precise structural control. The resulting architecture supports high water retention, biocompatibility, and mechanical stability.

SEM analysis of the HA-Ag-CBD hybrid gel (Figure 3d) reveals a smooth, continuous matrix with sparsely distributed bright nanoscale deposits. These high-contrast features correspond to the Ag-CBD complex embedded within the HA–poloxamer network. Particle sizes range from approximately 20 to 100 nm, suggesting a heterogeneous yet controlled dispersion. The uniform background indicates a homogenous hydrogel structure without prominent porous features, characteristic of a well-integrated HA–poloxamer system. The discrete distribution of the Ag-CBD complex confirms successful incorporation with minimal aggregation, maintaining matrix integrity and structural uniformity.

### 3.3. Dynamic Light-Scattering Analysis

The DLS analysis presented in the comparative plot (Figure 4) provides a detailed assessment of the PSD and hydrodynamic behavior of CBD, the Ag-CBD complex, the HA–poloxamer hydrogel matrix, and the newly formulated HA-Ag-CBD hybrid gel. Distinct differences in size profiles among the samples underscore their specific structural characteristics and varying degrees of colloidal stability.

The DLS comparison plot presented in Figure 4 illustrates the PSD of the base HA hydrogel, CBD, the Ag-CBD complex, and the final HA-Ag-CBD hybrid gel. The analysis provides insight into the colloidal behavior and dispersion stability of each system, highlighting the influence of component interactions on particle size heterogeneity.

The base hydrogel, composed of HA and the poloxamer, exhibited a primary particle size population within the nanometric range (~0.001–0.005 μm), suggesting a well-organized and uniformly hydrated polymeric network. This narrow size distribution is indicative of efficient physical crosslinking and structural homogeneity within the hydrogel matrix. The measured PDI (0.58) falls within the acceptable range for physically crosslinked hydrogels, confirming a consistent nanoscale formulation.

In contrast, CBD dispersed in an aqueous medium showed a broader PSD (~0.01–1 μm), reflecting a tendency to form micron-scale aggregates due to limited aqueous solubility. Despite a relatively low PDI (0.046), the presence of larger particles indicates a heterogeneous dispersion with reduced colloidal stability.

The Ag-CBD complex revealed a multimodal particle size profile, extending from ~0.01 to 1.2 μm, with multiple distinct peaks suggestive of hybrid aggregate formation. The corresponding PDI values (0.036, 1.473, and 0.131) point to a mixed population ranging from monodisperse to polydisperse domains. These results reflect dynamic interactions between AgNPs and CBD molecules, forming variably sized aggregates that confirm the successful formation of hybrid nanostructures.

The final HA-Ag-CBD hybrid gel exhibited a significantly refined and more homogeneous PSD, with a dominant peak in the low nanometer range (~0.001–0.01 μm). Compared to the individual and binary systems, this formulation demonstrated superior size uniformity and a notable reduction in aggregation. Although some higher PDI values (0.966, 52.686, and 7.077) were recorded, these were associated with minor secondary populations and did not impact the main nanometric fraction. The primary peak confirms the effective dispersion and stabilization of both the CBD and AgNPs within the HA–poloxamer matrix.

Overall, the DLS analysis demonstrates that the integration of the Ag-CBD complex into the HA hydrogel matrix results in a stable, nanoscale system with enhanced dispersion and minimized heterogeneity. This uniform particle profile is particularly advantageous for biomedical applications, where nanoscale dimensions and colloidal stability are essential for enhanced cellular uptake, improved bioavailability, and sustained release. The results confirm that the HA-Ag-CBD hybrid gel exhibits optimized physicochemical characteristics, making it a promising candidate for therapeutic topical delivery.

The DLS analysis demonstrates that the incorporation of the Ag-CBD complex into the HA–poloxamer matrix results in a well-dispersed nanoscale system with reduced particle aggregation and enhanced size uniformity. Compared to the individual and binary components, the final HA-Ag-CBD hybrid gel exhibits improved colloidal stability and homogeneity. The nanoscale PSD and consistent hydrodynamic behavior observed in the composite formulation reflect effective stabilization of the active components within the hydrogel network. These physicochemical characteristics are essential for biomedical applications, particularly in topical therapeutic delivery systems, where uniformity and stability influence performance, reproducibility, and safety.

### 3.4. Rheological Properties of the HA-Ag-CBD Hybrid Gel

The rheological behavior of the HA-Ag-CBD hybrid gel was systematically characterized to evaluate its suitability for topical or sprayable biomedical applications.

A temperature-controlled rheometry protocol was employed to monitor changes in viscosity, flow behavior, and structural stability across a physiologically relevant temperature range (16–36 °C). This approach allowed insights into the gel’s thermal responsiveness, flow stability, and shear-dependent behavior.

Figure 5 presents the rheograms of the HA-Ag-CBD hybrid gel measured across a range of temperatures, revealing a distinct temperature-dependent viscoelastic behavior.

At lower temperatures (16 °C and 22 °C), the gel displays Newtonian flow characteristics, as evidenced by the linear shear stress–shear rate relationships and excellent correlation coefficients (*R*^2^ values 0.9992 and 0.9990, respectively). These profiles suggest a low-resistance, predictable flow regime ideal for sprayability and ease of application, especially under ambient or refrigerated conditions. However, as the temperature rises to 28 °C and 36 °C, the flow behavior shifts markedly. The flow curves deviate from linearity, and a pronounced yield stress emerges, reflected by intercepts on the shear stress axis. This indicates a transition to non-Newtonian plastic behavior, further corroborated by decreasing *R*^2^ values (0.9779 at 28 °C and 0.8677 at 36 °C). These changes suggest the formation of internal structural interactions, likely due to thermal activation of polymer and NP interactions, promoting a cohesive gel state suitable for in situ film formation upon skin contact.

Figure 6 illustrates the temperature-dependent trends of viscosity and yield stress.

Below 22 °C, the gel exhibits negligible yield stress (<10 Pa) and maintains a low viscosity (<40 mPa·s), supporting a fluid-like consistency that facilitates uniform application over sensitive or irregular skin surfaces. Above 22 °C, both yield stress and viscosity rise significantly. Yield stress increases steeply to ~35 Pa at 26 °C and stabilizes around ~40 Pa between 28 °C and 36 °C, while viscosity peaks at 0.22 Pa·s at 28 °C. These findings define 22 °C as a functional transition point, beyond which the gel assumes a more structured, stable consistency. The presence of yield stress at these elevated temperatures implies the development of internal cohesion and network strength, which enhances topical retention and provides a stable physical barrier after application.

The establishment of measurable yield stress under these conditions suggests the development of internal network strength, promoting topical retention and physical barrier formation upon skin contact.

Figure 7 explores the effect of the shear rate on viscosity at different temperatures, simulating mechanical stresses encountered during spreading or movement.

At 16 °C and 22 °C, viscosity remains constant across shear rates, confirming Newtonian flow behavior. However, at 28 °C and 36 °C, viscosity becomes increasingly dependent on the shear rate. At low shear rates (<10 s^−1^), viscosity rises sharply, and threshold shear stresses reach up to 50 Pa, consistent with the plastic-like behavior indicated in previous figures. This growing resistance to deformation with temperature suggests that the gel responds mechanically to skin contact or manipulation by reinforcing its structural integrity, an advantageous property for film-forming dermal applications.

Figure 8a–c presents the gel’s response to ramp-up and ramp-down shear tests at 16 °C, 22 °C, and 36 °C, designed to evaluate time-dependent behavior such as thixotropy or rheopexy.

At all three temperatures, the ascending and descending flow curves overlap, demonstrating the absence of hysteresis. This consistency indicates that the gel structure remains stable under mechanical stress and recovers without permanent breakdown or thickening, a crucial property for predictable performance during repeated application or wear. Furthermore, the gel transitions from dilatant at 16 °C to Newtonian at 22 °C and plastic at 36 °C, highlighting its thermally adaptive flow behavior across physiologically relevant conditions.

Figure 9 identifies an exception to the time-independent flow behavior.

At 24 °C, the gel displays a significant hysteresis loop between the ramp-up and ramp-down flow curves, indicative of thixotropic behavior. The thixotropic index, calculated at 83.78%, reflects a pronounced breakdown and subsequent recovery of the internal structure under shear. This finding suggests that 24 °C represents a metastable zone where the gel microstructure is particularly responsive to mechanical stimuli. Although not representative of the broader temperature profile, this localized anomaly may offer opportunities for tuning delivery properties through careful thermal or mechanical conditioning.

Collectively, the rheological results demonstrate that the HA-Ag-CBD hybrid gel exhibits a sharp, reproducible transition centered at 22 °C. Below this threshold, the gel behaves as a low-viscosity Newtonian fluid, ideal for spraying and smooth application. Above this temperature, it transitions into a semi-solid plastic material with increased viscosity and yield stress, supporting enhanced skin adherence and barrier functionality. The isolated thixotropic behavior at 24 °C adds nuance to its rheological profile, suggesting responsiveness that could be harnessed for tailored release or application strategies. Altogether, these results confirm the gel’s promise as a thermoresponsive topical system, combining ease of administration with robust performance at body temperature.

### 3.5. In Vitro Release Study

The in vitro release profiles of CBD and the AgNPs were assessed to compare the release kinetics from the non-gelled Ag-CBD complex and the HA-Ag-CBD hydrogel, thereby elucidating the role of the hydrogel matrix in achieving sustained and controlled delivery of both active components. The corresponding cumulative release data are shown in Figure 10.

The Ag-CBD complex exhibited the rapid release of both CBD and AgNPs. The release kinetics followed a predominantly diffusion-controlled mechanism, as expected for a non-gelled dispersion. Release data fitted to the Korsmeyer–Peppas model yielded the following (Equations (2) and (3)):(2)$$CBD=\frac{M_{t}}{M_{\infty }}=(0.068\pm 0.003)t^{\left(0.62\pm 0.03\right)}\;(R^{2}=0.987)$$



(3)
$$AgNPs=\frac{M_{t}}{M_{\infty }}=(0.064\pm 0.003)t^{\left(0.60\pm 0.03\right)}\;(R^{2}=0.985)$$



The cumulative release of CBD increased from 5.12 ± 1.03% at 0.5 h to 95.34 ± 1.65% at 48 h, while AgNP release increased from 4.35 ± 0.95% to 92.15 ± 1.89% (Figure 10). The release profiles of CBD and the AgNPs were highly correlated (*r* = 0.97, *p* < 0.001), reflecting the stable co-dispersion of the active components. The Korsmeyer–Peppas exponents (n ≈ 0.60) confirm a Fickian diffusion-dominated release, consistent with the absence of a structured matrix to modulate transport.

In contrast, the HA–poloxamer hydrogel provided controlled and sustained release. The cumulative release of CBD increased from 2.87 ± 0.92% at 0.5 h to 90.12 ± 1.87% at 48 h, while AgNP release rose from 1.78 ± 0.76% to 85.67 ± 2.32% over the same period. Gel dissolution progressed from 3.65 ± 1.12% to 88.45 ± 2.10%, closely mirroring the release profiles. The release and dissolution data exhibited strong correlations for CBD (*r* = 0.96, *p* < 0.001) and for AgNPs (*r* = 0.95, *p* < 0.001), indicating that matrix erosion contributed to the release.

Korsmeyer–Peppas modeling confirmed anomalous transport (Equations (4) and (5)):(4)$$CBD=\frac{M_{t}}{M_{\infty }}=(0.045\pm 0.002)t^{\left(0.78\pm 0.02\right)}\;(R^{2}=0.994)$$



(5)
$$AgNPs=\frac{M_{t}}{M_{\infty }}=(0.041\pm 0.002)t^{\left(0.74\pm 0.03\right)}\;(R^{2}=0.991)$$



The higher exponents (n = 0.74–0.78) indicate that release is governed by both diffusion and hydrogel matrix relaxation/erosion, highlighting the hydrogel’s capacity for prolonged, synchronized delivery of CBD and AgNPs.

Notably, the HA–poloxamer hydrogel significantly modulated the initial burst release, reducing the rapid spike observed in the Ag-CBD complex. This controlled release behavior is critical for maintaining therapeutic concentrations over extended periods, minimizing cytotoxic peaks, and sustaining antimicrobial efficacy.

Overall, these results demonstrate that the HA-Ag-CBD hydrogel provides a superior delivery platform to that of the unstructured Ag-CBD complex, enabling precise temporal control of co-delivered active agents, which is essential for antimicrobial and cytotoxic applications.

### 3.6. Evaluation of Antimicrobial Activity

The antimicrobial efficacy of the CBD, colloidal Ag, their binary complex (Ag-CBD), and the newly formulated HA-Ag-CBD hybrid gel was systematically assessed against selected Gram-positive (*S. aureus*), Gram-negative (*E. coli*, *P. aeruginosa*), and fungal (*C. albicans*) pathogens. Antimicrobial performance was initially evaluated by agar disk diffusion, followed by MIC and MBC assays. Gentamicin (100 μg/mL) served as the positive control, while the HA gel matrix (10 μg/mL) was used as the negative control.

The comparative antimicrobial activities are summarized in Table 2 and Table 3.

Gentamicin was employed as a uniform positive control across all assays to ensure methodological consistency. Although it lacks intrinsic antifungal activity against *C. albicans*, its inclusion allowed standardized comparison, with future studies planned to incorporate dedicated antifungal control, such as fluconazole.

The HA–poloxamer gel matrix, evaluated independently against *S. aureus*, *E. coli*, *P. aeruginosa*, and *C. albicans*, produced no measurable IZs (0 mm) in the disk diffusion assay and exhibited MIC and MBC values exceeding 400 μg/mL in broth microdilution assays. These results confirm that the gel base itself lacks intrinsic antimicrobial activity, ensuring that the inhibitory effects observed in the HA-Ag-CBD formulations arise exclusively from the incorporated CBD, colloidal Ag, or their hybrid complex.

CBD demonstrated significant antimicrobial activity, with mean IZs of 9.07 ± 0.35 mm against *S. aureus* and 9.15 ± 0.46 mm against *C. albicans*. Colloidal Ag exhibited smaller IZs—6.32 ± 0.41 mm for *S. aureus*, 6.58 ± 0.37 mm for *C. albicans*, and 6.53 ± 0.44 mm for *E. coli*—highlighting its comparatively low activity, particularly against Gram-negative strains.

The Ag-CBD hybrid showed comparable or slightly enhanced IZs relative to CBD alone, e.g., 9.17 ± 0.49 mm for *S. aureus*, while incorporation into the HA–poloxamer hydrogel preserved antimicrobial activity, with IZs of 9.53 ± 0.30 mm for *S. aureus* and 9.27 ± 0.38 mm for *C. albicans*. No IZs were observed for *P. aeruginosa* for any formulation except gentamicin, consistent with its intrinsic resistance.

Notably, the HA-Ag-CBD formulation exhibited an unusually high MBC of 387.35 μg/mL against *E. coli*, while the MIC remained low (9.24 ± 0.48 μg/mL). This discrepancy reflects the inherent resistance of Gram-negative bacteria, whose lipopolysaccharide-rich outer membrane limits antimicrobial penetration, and may also indicate modulation of local release rates by the hydrogel, requiring higher concentrations to achieve bactericidal activity. Importantly, effective growth inhibition is still achieved at much lower concentrations. These data confirm that the HA-Ag-CBD system effectively inhibits bacterial growth while highlighting the necessity of evaluating both MIC and MBC to fully interpret antimicrobial efficacy.

In broth microdilution assays, CBD demonstrated the lowest MIC and MBC/MFC values among all tested samples, indicating strong bactericidal and fungicidal activity (MIC: 1.74 ± 0.25 μg/mL for *S. aureus*, 1.98 ± 0.31 μg/mL for *C. albicans*; MBC/MFC: 2.15 ± 0.34 μg/mL and 2.57 ± 0.29 μg/mL, respectively). Colloidal Ag exhibited higher MICs (e.g., 2.67 ± 0.36 μg/mL for *S. aureus*, 3.43 ± 0.38 μg/mL for *E. coli*) and MBCs up to 4.25 ± 0.43 μg/mL for *E. coli*. The Ag-CBD hybrid retained substantial antimicrobial potency (MIC: 2.06 ± 0.29 μg/mL for *S. aureus*, 2.35 ± 0.33 μg/mL for *C. albicans*), with MBC/MFC values similar to those of CBD alone, consistent with additive effects, and hydrogel encapsulation did not compromise activity. HA-Ag-CBD MICs were comparable: 2.18 ± 0.27 μg/mL for *S. aureus* and 2.45 ± 0.30 μg/mL for *C. albicans*.

All formulations exhibited MIC and MBC values > 8 μg/mL against *P. aeruginosa*, confirming resistance. The absence of measurable IZs and MIC/MBC values for *P. aeruginosa* reflects its well-known intrinsic resistance. This Gram-negative pathogen possesses a highly impermeable outer membrane, active efflux pumps, and biofilm-forming capabilities, all of which limit the uptake and efficacy of antimicrobial agents, including CBD, colloidal Ag, and their hybrid formulations. These features account for the lack of observable activity under the current assay conditions.

Statistical analysis confirmed that CBD significantly outperformed colloidal Ag across all susceptible strains (*p* < 0.01), while differences among CBD, Ag-CBD, and HA-Ag-CBD were not significant (*p* > 0.05), underscoring the preservation of CBD’s antimicrobial potency in the hybrid and hydrogel systems.

These findings are consistent with the existing literature, which recognizes CBD’s pronounced antimicrobial effects, particularly against Gram-positive bacteria and fungi, and confirms the moderate but measurable efficacy of colloidal Ag under similar experimental conditions. These MIC and MBC/MFC values align with previous reports, reinforcing the observed efficacy of CBD and colloidal Ag under in vitro conditions and validating the reliability of the findings.

Overall, these results demonstrate that both CBD and colloidal Ag possess antimicrobial properties, with CBD displaying superior efficacy, particularly against *S. aureus* and *C. albicans*. Incorporation of these agents into the Ag-CBD hybrid and subsequent integration into the HA–poloxamer hydrogel maintained their bioactivity without loss of efficacy, consistent with the existing literature [46,47,48,49,50,51], supporting their potential for topical antimicrobial applications.

### 3.7. Cell Viability Assay

The results, presented in Figure 11 and Figure 12, inform the selection of optimized concentrations for further formulation refinement and preclinical development in skin-focused biomedical applications.

Cytotoxicity assays on NHDFs revealed that both concentration (*p* < 0.0001) and incubation time (*p* < 0.0001) significantly influenced cell viability across all tested compounds. A significant interaction between concentration and time (*p* < 0.001) indicated that cytotoxic effects were enhanced synergistically with increasing dose and exposure duration. Tukey’s HSD post hoc analysis demonstrated that at concentrations ≥50 μg/mL, all active formulations except the HA–poloxamer gel matrix significantly reduced cell viability compared to untreated controls (*p* < 0.01), with notable differences among formulations.

CBD exhibited moderate, dose-dependent cytotoxicity, maintaining > 95% viability at 25 μg/mL across all timepoints, while at 100 μg/mL, viability declined to 65.2 ± 3.5%, 55.6 ± 4.1%, and 48.7 ± 4.3% at 24, 48, and 72 h, respectively. Colloidal Ag displayed slightly stronger cytotoxicity at higher concentrations, with 40.6 ± 4.8% viability at 72 h. Differences between CBD and Ag were not statistically significant (*p* > 0.05), consistent with the literature reporting mild-to-moderate dermal fibroblast cytotoxicity for both compounds at elevated doses [52,53,54,55,56,57].

The Ag-CBD complex exhibited the highest cytotoxicity among the active agents, with cell viability of 52.4 ± 4.0%, 43.6 ± 4.7%, and 35.2 ± 5.0% at 100 μg/mL across 24, 48, and 72 h, respectively. These values were significantly lower than those for CBD or Ag alone (*p* < 0.05), suggesting a synergistic cytotoxic effect likely mediated by CBD-facilitated AgNP uptake and intracellular Ag^+^ release.

Incorporation of the Ag-CBD complex into the HA–poloxamer hydrogel significantly mitigated cytotoxicity, maintaining viability above 75% at concentrations ≤ 50 μg/mL. At 100 μg/mL, the HA-Ag-CBD hydrogel yielded 61.5 ± 3.6%, 52.3 ± 4.3%, and 45.8 ± 4.6% viability at 24, 48, and 72 h, respectively, significantly higher than the Ag-CBD complex alone (*p* < 0.05). The HA–poloxamer matrix alone preserved > 92% viability across all concentrations and timepoints, confirming its inertness and excellent biocompatibility.

Dose- and time-dependent IC_50_ values were derived using nonlinear regression from triplicate measurements. CBD IC_50_ decreased from 112.3 ± 5.2 μg/mL (24 h) to 82.1 ± 4.5 μg/mL (72 h), while colloidal Ag ranged from 98.5 ± 5.0 μg/mL to 67.8 ± 4.2 μg/mL, reflecting slightly higher cytotoxicity. The Ag-CBD complex exhibited lower IC_50_ values (85.6 ± 4.9 μg/mL at 24 h; 58.4 ± 4.0 μg/mL at 72 h; *p* < 0.01), consistent with synergistic cytotoxicity. Incorporation into the hydrogel significantly increased IC_50_ values (105.7 ± 5.1 μg/mL at 24 h; 76.3 ± 4.4 μg/mL at 72 h; *p* < 0.01), confirming the protective effect of the HA–poloxamer matrix. The HA–poloxamer exhibited IC_50_ values >200 μg/mL at all timepoints, with no difference from untreated controls (*p* > 0.05).

Overall, the HA-Ag-CBD hydrogel effectively mitigates the intrinsic cytotoxicity of the Ag-CBD complex while maintaining fibroblast viability above 70% at therapeutically relevant concentrations, fulfilling International Organization for Standardization (ISO) 10993-5:2009 [58] and European Medicines Agency (EMA) [59] guidelines for topical applications. These findings establish a clear therapeutic window and highlight the hydrogel’s potential as a safe, biocompatible delivery system for dermal applications.

## 4. Discussion

The reproducible low-temperature preparation method, confirmed across independent batches, underpins the consistent nanometric particle distribution, rheological properties, and homogeneous dispersion of the HA-Ag-CBD hydrogel, ensuring reliability and translational potential for topical applications.

FTIR analysis (Figure 2b,c) provides clear evidence for the successful synthesis and molecular integration of the HA-Ag-CBD hybrid gel via a cold-processing approach. The spectrum of colloidal Ag displays characteristic signals from the stabilizing agent, essential for maintaining NP stability and surface reactivity, thereby facilitating subsequent interactions. In the Ag-CBD complex, distinct spectral shifts, particularly in the carboxylate and aromatic stretching regions, indicate chemisorptive interactions between CBD’s phenolic hydroxyls and carboxyl moieties and the Ag surface. This stable surface anchoring is critical for preserving the structural integrity and therapeutic functionality of CBD under low-temperature formulation conditions.

The FTIR spectrum of the final HA-Ag-CBD hydrogel confirms cohesive integration of all components—disappearance of the free carboxylic acid band at 1748 cm^−1^ and the appearance of coordinated carboxylate peaks at 1646 cm^−1^ and 1457 cm^−1^ suggest strong hydrogen bonding and electrostatic interactions between HA and the embedded Ag-CBD complex. Retention of characteristic fingerprint bands of both CBD and Ag indicates that the cold-processing method preserves chemical identity and functional integrity.

The observed stability and uniformity of the HA-Ag-CBD hydrogel can be attributed to specific interactions between HA and colloidal Ag. HA’s carboxyl and hydroxyl groups likely coordinate with AgNP surfaces, stabilizing the NPs and preventing aggregation within the polymeric network. These interactions maintain the antimicrobial efficacy of the Ag-CBD complex while enabling controlled release of active agents, mitigating cytotoxic effects on fibroblasts. Consequently, HA functions both as a biocompatible scaffold and stabilizing agent, facilitating sustained delivery while maintaining cell viability above ISO 10993-5:2009 [58] thresholds. This mechanistic insight underscores HA’s pivotal role in enhancing formulation performance and ensuring safe, effective topical therapeutics.

SEM analysis provides complementary structural validation. CBD exhibits an irregular, semi-amorphous nanoscale morphology, consistent with its encapsulated, oil-based nature. The Ag-CBD complex forms core–shell structures, with 50–150 nm AgNP cores enveloped by fibrous CBD layers, reflecting strong surface interactions and stabilization. The HA–poloxamer hydrogel presents a porous, semi-crystalline network, promoting uniform dispersion, water retention, and mechanical integrity. Incorporation of the Ag-CBD complex into the HA matrix results in a smooth, continuous hydrogel with discrete nanoscale deposits (20–100 nm), demonstrating controlled dispersion, minimal aggregation, and well-preserved architecture. These observations provide a structural basis for the hydrogel’s reproducible nanometric fraction and its rheological properties, highlighting a coherent structure–function relationship that underpins mechanical stability, thermoresponsive behavior, and uniform, sustained topical delivery.

DLS analysis further corroborates these findings, showing a dominant nanometric fraction (~0.001–0.01 μm) and confirming effective stabilization of CBD and the AgNPs within the HA–poloxamer matrix. Occasional high PDI values (up to 52.686) correspond to minor secondary populations and do not compromise the primary nanometric fraction that governs therapeutic performance. Nanoscale uniformity and colloidal stability are essential for predictable bioavailability, cellular uptake, and sustained release in topical applications. Minor aggregation can be mitigated through process optimization, including controlled mixing, ultrasonication, and precise polymer-to-particle ratios, ensuring batch-to-batch reproducibility.

Collectively, FTIR, SEM, and DLS analyses confirm that the HA-Ag-CBD hybrid gel exhibits homogeneous dispersion, preserved particle size, and structural integrity, supporting its reliability and translational potential. While the quantitative release of CBD and AgNPs was not directly measured, the observed dose-dependent cytotoxicity demonstrates effective bioactive delivery. Future work will focus on detailed release kinetics to directly correlate agent release with biological response, further substantiating the hydrogel as a multifunctional platform for topical therapeutic applications.

The rheological profile of the HA-Ag-CBD hybrid gel underscores its potential as a next-generation topical spray for epidermal lesions.

Below 22 °C, the gel displays Newtonian-like flow with low viscosity (<40 mPa·s) and negligible yield stress, properties that are ideal for sprayable formulations, as they enable smooth passage through nozzles, uniform distribution, and dose reproducibility (*R*^2^ > 0.999).

The fluidity under ambient conditions simplifies both clinical and home use, preventing nozzle clogging or uneven application. These characteristics likely arise from weak intermolecular interactions, dominated by transient hydrogen bonding and limited crosslinking among HA, CBD, and colloidal Ag. Once applied to skin (32–36 °C), the gel undergoes a thermally triggered transition into a plastic, non-Newtonian state with measurable yield stress (~40 Pa) and higher viscosity (~0.22 Pa·s at 28 °C). This in situ gelation, mediated by hydrophobic interactions, hydrogen bonding, and ionic associations, ensures a semi-solid layer that resists runoff, prolongs residence time, and supports sustained local delivery.

Interestingly, a reproducible rheological anomaly was observed at 24 °C, where the gel exhibited a pronounced thixotropic hysteresis loop (index ≈ 83.8%). This response, consistently replicated across trials, suggests a formulation-intrinsic property rather than an artifact. We hypothesize that it reflects a metastable microstructural state involving reversible HA chain rearrangements coupled with Ag-CBD coordination dynamics. Comparable transitional thixotropy has been reported in poloxamer-based gels, pointing to a possible compositional origin [60,61,62].

While the precise mechanism requires further study, such a shear-sensitive transition may even be advantageous for applications requiring mechanical stimulation, such as massage-assisted delivery or micro-needling.

Functionally, the HA-Ag-CBD gel exemplifies a smart, phase-responsive system that seamlessly transitions from a low-viscosity, sprayable liquid to a cohesive semi-solid matrix at skin temperature. Unlike conventional hydrogels or creams, which often compromise either mechanical integrity or ease of use, this formulation balances both while integrating therapeutic actives—HA (hydration, tissue repair), CBD (anti-inflammatory, antioxidant), and colloidal Ag (antimicrobial)—directly into its matrix. Compared to poloxamer or synthetic poly(*N*-isopropylacrylamide) (PNIPAM) hydrogels, the hybrid system offers superior biocompatibility and therapeutic payload delivery without additional carriers or excipients. Its stability under dynamic shear further supports reliability in wound care, burn treatment, and dermal DDSs, where patient movement, perspiration, or friction may otherwise compromise performance.

Collectively, these rheological results validate the HA-Ag-CBD hybrid gel as a multifunctional, thermoresponsive delivery platform. Its controlled sprayability at room temperature and solidification at physiological temperature enable both user-friendly administration and prolonged local activity. The isolated thixotropic behavior at 24 °C represents a unique design feature that merits deeper investigation but does not compromise the formulation’s robustness. Future work should focus on elucidating the molecular drivers of this transition, optimizing long-term stability, and confirming therapeutic efficacy in preclinical and clinical models.

The release profiles highlight a fundamental mechanistic distinction between the unstructured Ag-CBD complex and the HA-Ag-CBD hydrogel. The Ag-CBD complex exhibited a burst-type release, dominated by rapid desorption of surface-bound actives, producing transient supra-therapeutic concentrations but with poor temporal control. In contrast, the hydrogel network governed release through a dual mechanism involving matrix–drug interactions, polymer relaxation, and gradual hydrolytic erosion. This composite mechanism enabled synchronized, near zero-order release of both CBD and AgNPs, thereby avoiding premature depletion and mitigating toxic concentration spikes. Modulation of release kinetics thus provides the mechanistic foundation for the observed divergence in biological outcomes.

The burst release from the Ag-CBD complex correlated with strong but short-lived antimicrobial activity, with inhibition against *S. aureus* and *C. albicans* declining sharply after 24 h. Simultaneously, the high initial concentrations translated into elevated fibroblast cytotoxicity at early exposure points, consistent with oxidative and mitochondrial stress induced by Ag^+^ and CBD overload. In contrast, the HA-Ag-CBD hydrogel achieved a slower release rate (*k* reduced by ~35%), with diffusional exponents (n > 0.70) indicative of anomalous transport governed by both diffusion and erosion. This controlled profile sustained antimicrobial efficacy for at least 72 h, maintaining >80% of the initial IZ while significantly reducing cytotoxicity through attenuation of peak exposures.

Statistical analyses reinforced these mechanistic insights—cumulative release was strongly correlated with antimicrobial persistence (*r* = 0.91, *p* < 0.001), while burst release was inversely correlated with NHDF viability (*r* = −0.87, *p* < 0.01). These findings confirm that hydrogel encapsulation fundamentally reshapes the therapeutic profile of the Ag-CBD complex, transforming it from a system characterized by transient potency and host cytotoxicity into one with sustained antimicrobial activity and improved biocompatibility.

The antimicrobial activity observed in this study highlights the potent efficacy of CBD against key pathogenic strains, particularly *S. aureus* and *C. albicans*. Gentamicin was used as a single positive control across all microorganisms to maintain methodological consistency and enable direct comparison of IZ data between bacterial and fungal assays. Although gentamicin lacks intrinsic antifungal activity, *C. albicans* was included primarily for comparative assessment rather than benchmarking against a standard antifungal. We acknowledge this as a limitation and note that future studies will incorporate a dedicated antifungal comparator, such as fluconazole, to enable direct potency evaluations.

The absence of antimicrobial activity from the HA–poloxamer gel matrix underscores its biological inertness as a delivery vehicle. This ensures that the potent inhibitory effects observed for the HA-Ag-CBD hybrid gel are solely attributable to the active components rather than the hydrogel matrix. The findings validate the HA–poloxamer gel as a safe, non-interfering carrier capable of preserving the bioactivity of encapsulated agents, highlighting its suitability for use in multifunctional antimicrobial and wound care formulations. CBD exhibited significant IZs and low MIC/MBC values, consistent with its known mechanisms, including disruption of microbial membranes, inhibition of biofilm formation, and interference with intracellular signaling pathways. These multifaceted modes of action likely contribute to the enhanced susceptibility of Gram-positive bacteria, and fungal strains to CBD, aligning well with the literature [46,47,48,49,50].

Colloidal Ag demonstrated moderate antimicrobial activity, with smaller IZs and higher MIC/MBC values, in agreement with its primary mode of action via Ag^+^-induced oxidative stress and disrupting cellular function in bacteria and fungi [51].

The HA–poloxamer gel matrix showed no antimicrobial activity (0 mm IZ, MIC/MBC > 400 μg/mL), confirming it is biologically inert and does not interfere with the bioactivity of loaded agents. Combined with cytotoxicity data showing high NHDF viability in the presence of the blank gel, these findings validate the hydrogel as a safe, biocompatible delivery platform.

Gram-negative bacteria, including *E. coli* and *P. aeruginosa*, exhibited lower susceptibility, reflecting their complex outer membrane structures, efflux pumps, and biofilm-forming capabilities. The absence of measurable IZs and MIC/MBC values for *P. aeruginosa* reflects its well-known intrinsic resistance. This Gram-negative pathogen possesses a highly impermeable outer membrane, active efflux pumps, and biofilm-forming capabilities, all of which limit the uptake and efficacy of antimicrobial agents, including CBD, colloidal Ag, and their hybrid formulations. These features account for the lack of observable activity under the current assay conditions.

While the Ag-CBD complex demonstrated slightly enhanced inhibition compared to CBD alone, the effect was primarily additive rather than fully synergistic, suggesting that the combination of CBD and AgNPs improves microbial susceptibility through complementary modes of action, such as enhanced membrane permeabilization and intracellular Ag^+^ delivery, without introducing a dramatically new mechanism.

The incorporation of the Ag-CBD complex into the HA gel (HA–poloxamer) matrix did not compromise its antimicrobial efficacy, as evidenced by statistically comparable IZs and MIC/MBC values between the Ag-CBD and HA-Ag-CBD formulations. This indicates that the hydrogel functions effectively as a biocompatible, sustained-release delivery platform, preserving the bioactivity of the encapsulated agents. The unusually high MBC of HA-Ag-CBD against *E. coli* underscores the differential susceptibility of Gram-negative bacteria compared to Gram-positive strains. Structural defenses, including a lipopolysaccharide-rich outer membrane and active efflux systems, impede bactericidal action, while hydrogel encapsulation may influence the release kinetics of the active agents.

A key observation in antimicrobial testing was the large disparity between MIC and MBC values for *E. coli* with the HA-Ag-CBD gel (MIC: 9.24 μg/mL; MBC: 387.35 μg/mL). This >40-fold difference indicates primarily bacteriostatic activity against this Gram-negative strain, reflecting its inherent structural resistance, including the lipopolysaccharide-rich outer membrane that limits penetration of antimicrobial agents. The hydrogel matrix may further modulate local release, contributing to higher bactericidal thresholds.

In contrast, Gram-positive *S. aureus* and the fungal strain *C. albicans* exhibited low MIC and MBC values, consistent with bactericidal activity. These findings emphasize that antimicrobial effects must be interpreted in the context of microbial physiology and structural defenses, and they validate the HA-Ag-CBD hydrogel as an effective in vitro antimicrobial platform while highlighting strain-specific differences in response.

CBD demonstrated statistically superior antimicrobial activity to colloidal Ag across all susceptible strains, reinforcing its potential as a broad-spectrum antimicrobial agent. The lack of significant differences in efficacy among the CBD, Ag-CBD, and HA-Ag-CBD formulations indicates that neither functionalization with Ag nor hydrogel encapsulation compromises CBD’s bioactivity. These results underscore the feasibility of integrating phytochemicals and nanomaterials into multifunctional delivery systems without diminishing therapeutic performance.

Collectively, the findings confirm that CBD, alone or in combination with colloidal Ag, can be effectively delivered via biocompatible hydrogels, achieving potent antimicrobial effects while addressing challenges such as solubility, stability, and controlled release. This supports the development of next-generation wound care materials and topical formulations with enhanced efficacy, consistent with the existing literature and demonstrating reproducible and reliable antimicrobial performance.

Cytotoxicity profiles of CBD, colloidal Ag, the Ag-CBD complex, and the HA-Ag-CBD hybrid gel were systematically evaluated in NHDFs, revealing significant effects of both sample type and exposure duration on cell viability. The observed dose- and time-dependent cytotoxicity provides a robust in vitro assessment of fibroblast compatibility despite IC_50_ values being derived from a relatively narrow concentration range (25–100 μg/mL). Triplicate measurements and well-defined concentration–response trends enabled reliable nonlinear regression analyses, offering meaningful comparative insights across formulations.

The IC_50_ values reported here are supported by the quantitative in vitro release profiles of CBD and the AgNPs from both the Ag-CBD complex and the HA-Ag-CBD hydrogel, enabling a direct correlation between released concentrations and the observed cytotoxic effects. These data provide a mechanistic basis for the hydrogel’s controlled, synchronized delivery of active components and substantiate its therapeutic performance in vitro.

Structural characterization by FTIR, SEM, and DLS confirmed successful incorporation of active components, supporting the reliability of the formulation and indicating effective delivery of bioactive agents from the hydrogel to the cells. CBD alone exhibited moderate, time-dependent cytotoxicity (IC_50_ ≈ 82 μg/mL at 72 h), consistent with mitochondrial and reactive oxygen species (ROS)-mediated mechanisms [55,56,57], whereas colloidal Ag showed stronger cytotoxicity (IC_50_ < 70 μg/mL at 72 h), in line with documented oxidative membrane damage and apoptosis induction [52,53,54]. The Ag-CBD complex demonstrated significantly enhanced cytotoxicity (*p* < 0.01 vs. individual agents), suggesting a synergistic effect likely mediated by CBD-facilitated AgNP uptake and intracellular Ag^+^ release.

Importantly, incorporation of the Ag-CBD complex into the HA–poloxamer hydrogel significantly mitigated the cytotoxic effects (IC_50_: 105.7 ± 5.1 μg/mL at 24 h; 76.3 ± 4.4 μg/mL at 72 h; *p* < 0.01), supporting the hypothesis that the hydrogel matrix modulates active release and reduces direct cellular exposure. The HA–poloxamer gel alone exhibited negligible cytotoxicity (IC_50_ > 200 μg/mL at all timepoints; *p* > 0.05 vs. control), confirming its inertness as a delivery platform.

At therapeutically relevant concentrations (≤50 μg/mL), the HA-Ag-CBD hybrid gel maintained NHDF viability above 75% over 72 h, meeting ISO 10993-5:2009 [58] and EMA [59] guidelines for topical safety. Moderate cytotoxicity observed at higher concentrations (≥75 μg/mL) exceeds typical therapeutic doses and is unlikely to be encountered in practical applications.

Collectively, these findings define a safe therapeutic window, confirm that hydrogel encapsulation preserves cell viability, and demonstrate effective delivery of active agents.

The integration of release kinetics with antimicrobial and cytotoxicity data highlights the HA-Ag-CBD hydrogel as a superior therapeutic platform to the unstructured Ag-CBD complex. By mitigating the burst release effect, prolonging antimicrobial activity, and significantly reducing fibroblast cytotoxicity, the new hybrid hydrogel achieves an improved therapeutic index. This represents a rational design strategy that balances potency with safety, addressing a critical challenge in multifunctional wound care systems.

The HA-Ag-CBD system therefore achieves a favorable balance between antimicrobial efficacy and biocompatibility, supporting its potential as a safe and efficient topical formulation for wound care and dermal applications.

## 5. Conclusions

This study successfully developed and characterized a novel thermoresponsive HA-Ag-CBD hybrid gel for topical antimicrobial applications. The formulation exhibited a physiologically favorable pH and gelation temperature, along with rheological features such as thixotropy and shear-thinning behavior, ensuring ease of application, spatial uniformity, and stability under dynamic skin conditions. These characteristics allow the gel to function as a sprayable, phase-responsive system that transitions into a cohesive semi-solid upon contact with skin, potentially prolonging residence time and supporting the sustained release of active agents. Importantly, release studies confirmed that hydrogel encapsulation suppressed burst release and enabled the sustained, controlled delivery of both CBD and AgNPs, directly supporting the observed balance between antimicrobial efficacy and cytocompatibility. Comprehensive in vitro biological evaluation demonstrated that the HA-Ag-CBD hybrid gel retains the antimicrobial activity of its active constituents. CBD consistently showed superior efficacy to colloidal Ag, while the Ag-CBD complex exhibited modestly enhanced activity, indicative of additive or potentially synergistic effects mediated by increased membrane permeabilization or intracellular Ag^+^ uptake. Cytotoxicity assays revealed that although the Ag-CBD complex displayed higher cellular toxicity, encapsulation within the HA–poloxamer matrix significantly mitigated these effects. At therapeutically relevant concentrations, the HA-Ag-CBD hybrid gel maintained fibroblast viability above the ISO 10993-5:2009 threshold, confirming in vitro biocompatibility.

While these results establish the HA-Ag-CBD gel as a promising in vitro platform combining antimicrobial efficacy with cytocompatibility, it is important to note that no in vivo studies have been conducted. Therefore, conclusions regarding multifunctional wound therapy or clinical applicability remain preliminary. Future work will focus on in vivo efficacy and safety evaluations, detailed skin permeation studies, and further formulation optimization to support the translational development of the HA-Ag-CBD hydrogel as a multifunctional therapeutic platform. The integration of the current in vitro release data provides a strong foundation for these next-stage preclinical studies.

Collectively, these findings provide a robust foundation for further development of thermoresponsive hydrogel-based delivery systems integrating natural and inorganic antimicrobial agents, with the potential to inform next-generation topical therapeutics, contingent upon comprehensive in vivo validation.

## Figures and Tables

**Figure 1 pharmaceutics-17-01189-f001:**
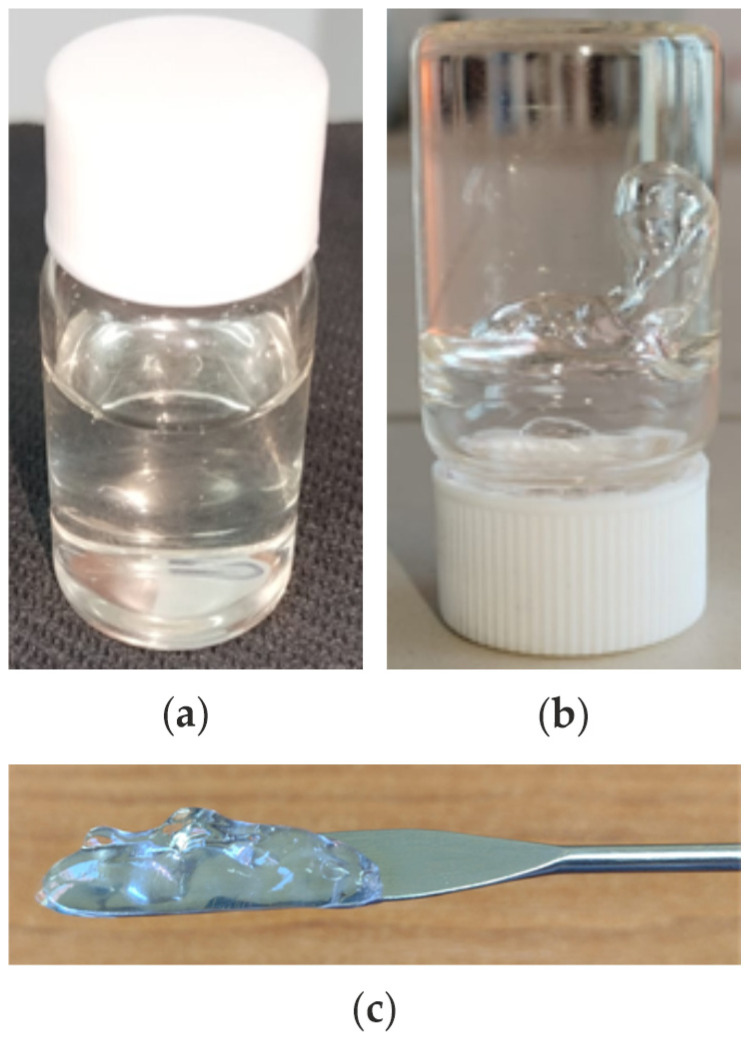
Macroscopic appearance of the HA-Ag-CBD hydrogel in liquid (**a**) and gelled (**b**,**c**) states: (**a**) hydrogel in its liquid form prior to gelation, demonstrating ease of handling and injectability; (**b**,**c**) hydrogel after sol-to-gel transition at 37 °C, showing the stable, self-supporting gel structure suitable for topical applications. Ag: Silver; CBD: Cannabidiol; HA: Hyaluronic acid.

**Figure 2 pharmaceutics-17-01189-f002:**
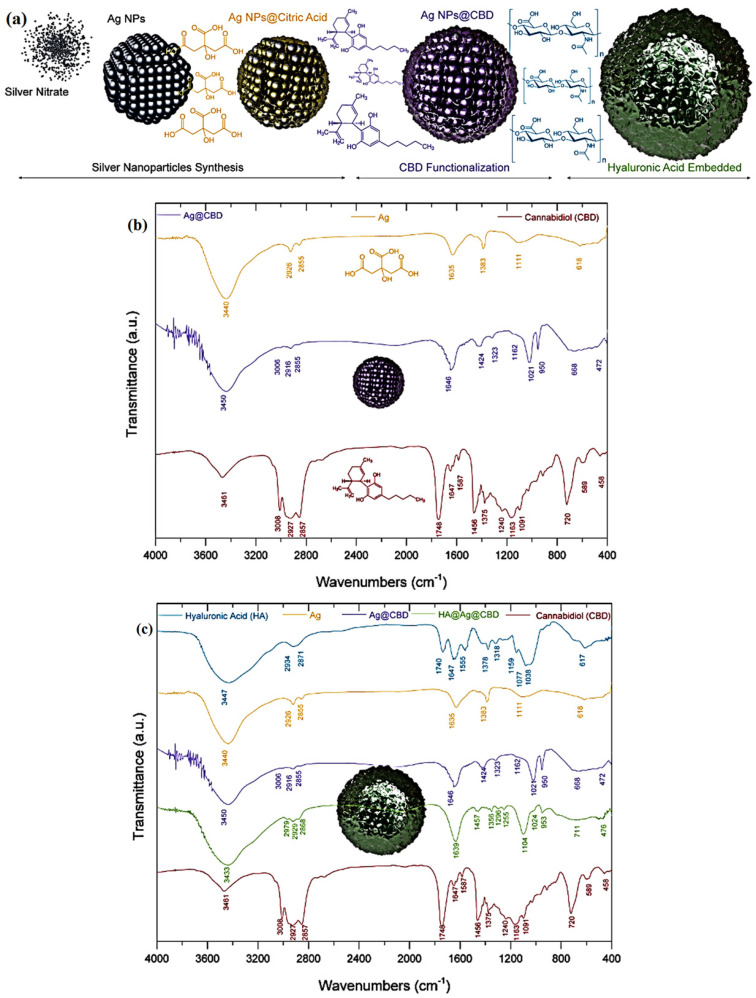
Schematic representation of the experimental protocol for the preparation of the new hybrid gel formulation: (**a**) superimposed FTIR spectra (from top to bottom) of colloidal Ag, Ag-CBD, and CBD, (**b**) and superimposed FTIR spectra (from top to bottom) of HA gel (HA–poloxamer) matrix, colloidal Ag, CBD, Ag-CBD, and HA-Ag-CBD, (**c**) highlighting the successful process of obtaining the hybrid nanocomposite. Ag: Silver; CBD: Cannabidiol; FTIR: Fourier-transform infrared; HA: Hyaluronic acid.

**Figure 3 pharmaceutics-17-01189-f003:**
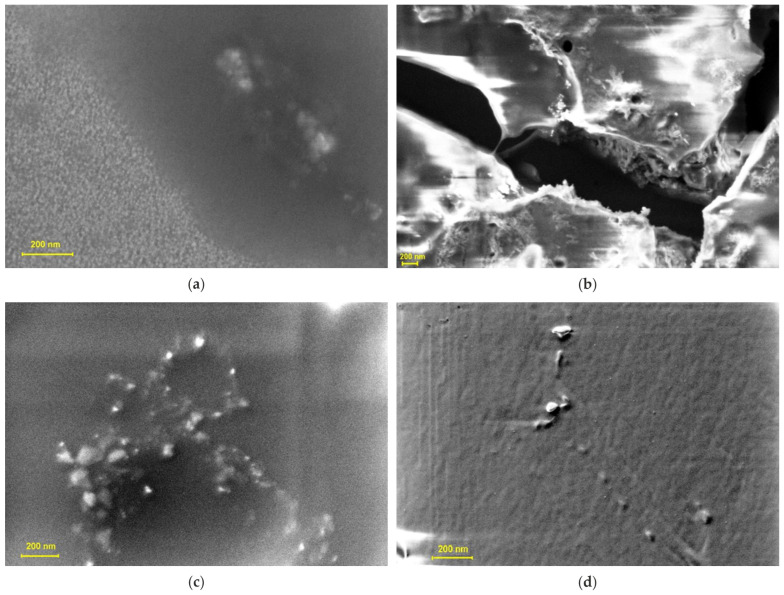
SEM micrograph of CBD (**a**), Ag-CBD (**b**), the HA gel (HA–poloxamer) matrix (**c**), and the HA-Ag-CBD hybrid gel (**d**). SEM: Scanning electron microscopy.

**Figure 4 pharmaceutics-17-01189-f004:**
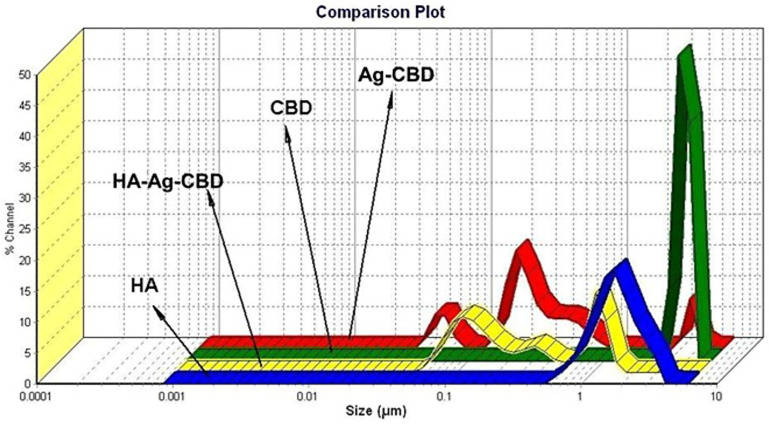
DLS pattern of CBD, Ag-CBD, the HA hydrogel, and the HA-Ag-CBD hybrid gel. DLS: Dynamic light scattering.

**Figure 5 pharmaceutics-17-01189-f005:**
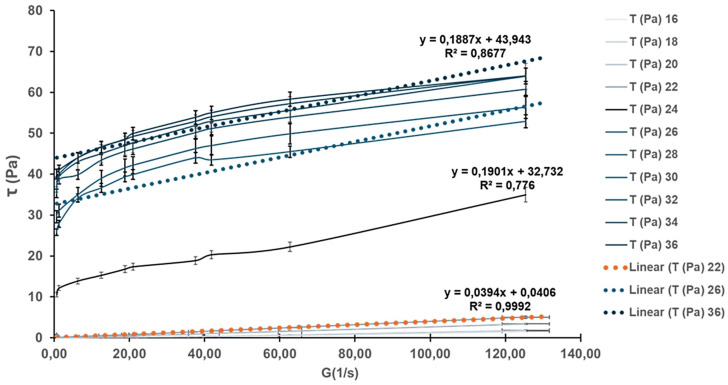
Rheogram of the HA-Ag-CBD hybrid gel for different temperatures: ramp-up curve. The viscosity of the fluid, represented by the slope of the graph curve, drastically changes as the temperature exceeds the threshold of 22 °C. Values are expressed as the mean ± SD (n = 3). SD: Standard deviation.

**Figure 6 pharmaceutics-17-01189-f006:**
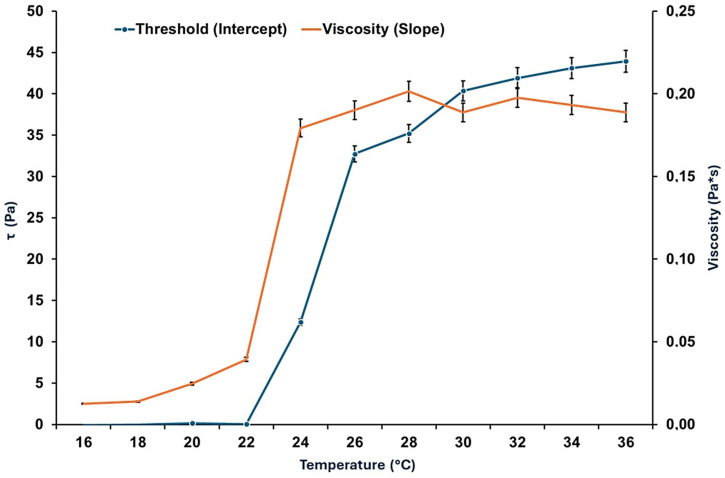
Threshold shear tension and viscosity vs. temperature. The intercept of each plot, corresponding to the threshold shear tension, from the above graph was plotted against temperature. The same representation was created for the average slope of each plot, corresponding to an average viscosity value. Values are expressed as the mean ± SD (n = 3).

**Figure 7 pharmaceutics-17-01189-f007:**
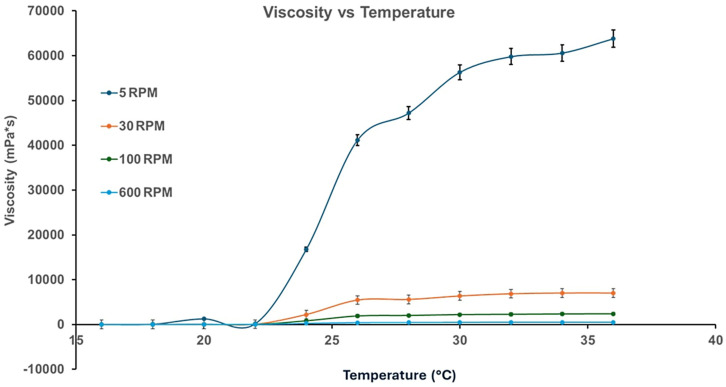
Graphical representation of viscosity vs. temperature for the HA-Ag-CBD hybrid gel, with a comparison of different revolutions per minute (RPM) plots. Values are expressed as the mean ± SD (n = 3).

**Figure 8 pharmaceutics-17-01189-f008:**
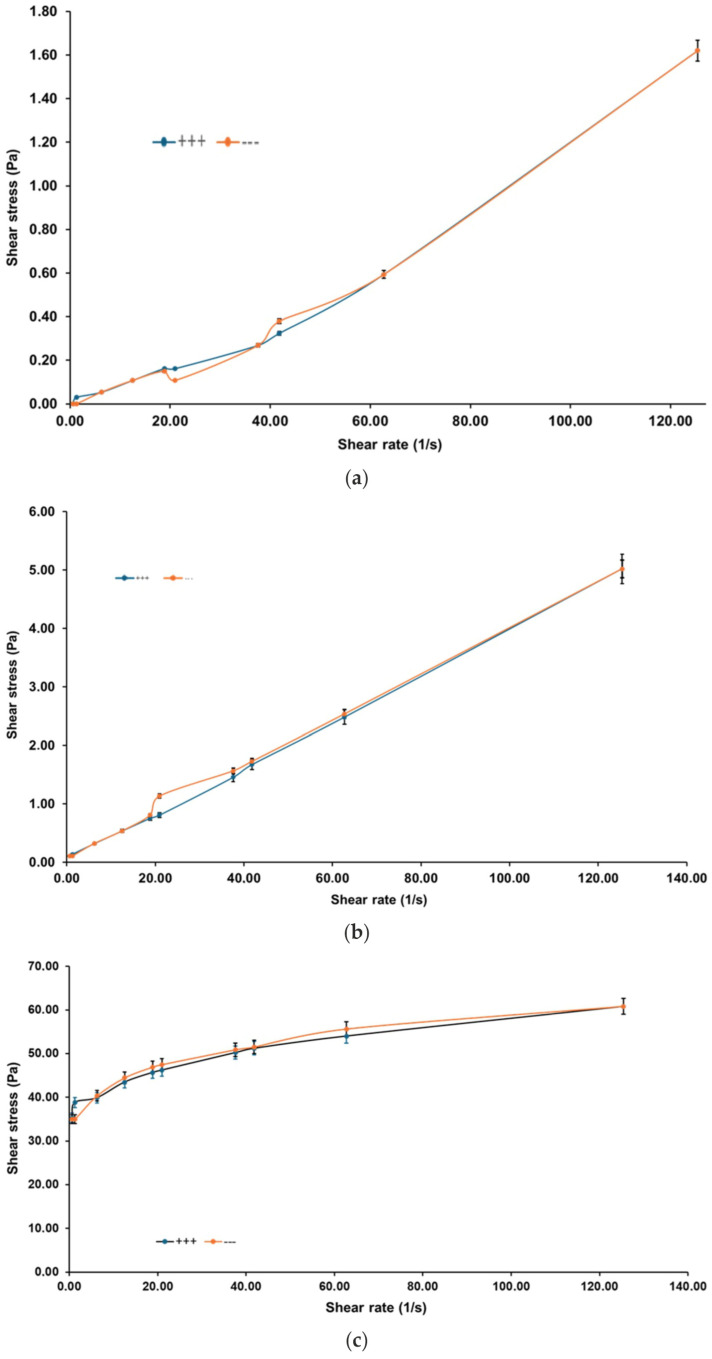
Rheograms for 16 °C (**a**), 22 °C (**b**), and 36 °C (**c**), showing the change in liquid flow as the temperature increases. The blue line is the ramp-up curve, and the red line is the ramp-down curve. No thixotropic or rheopectic behavior is observed. Values are expressed as the mean ± SD (n = 3).

**Figure 9 pharmaceutics-17-01189-f009:**
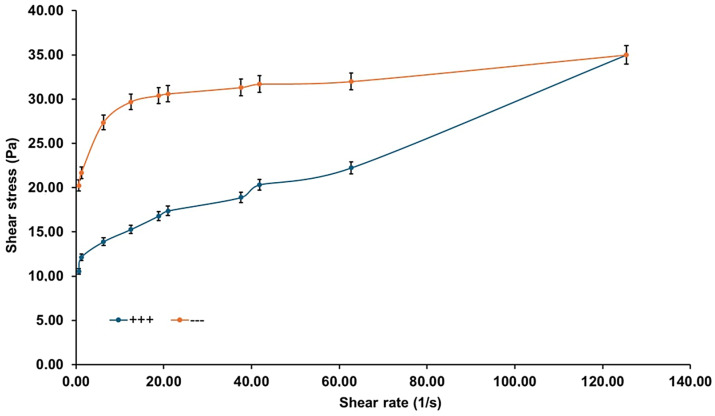
Rheogram registered at 24 °C. The sample is exhibiting a large thixotropic area. Values are expressed as the mean ± SD (n = 3).

**Figure 10 pharmaceutics-17-01189-f010:**
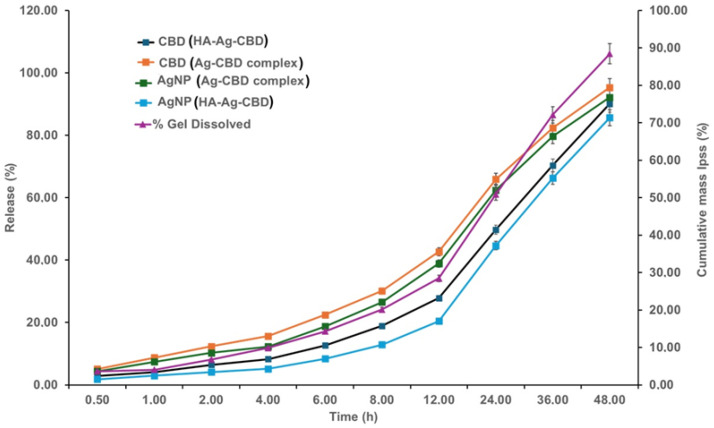
In vitro CBD and AgNP release from the Ag-CBD complex and HA-Ag-CBD hydrogel in PBS (pH 7.4, 5% v/v ethanol) at 37 °C as a function of time, and gel dissolution (% mass loss) over time for the hydrogel. Data represent the mean ± SD (n = 3). PBS: Phosphate-buffered saline.

**Figure 11 pharmaceutics-17-01189-f011:**
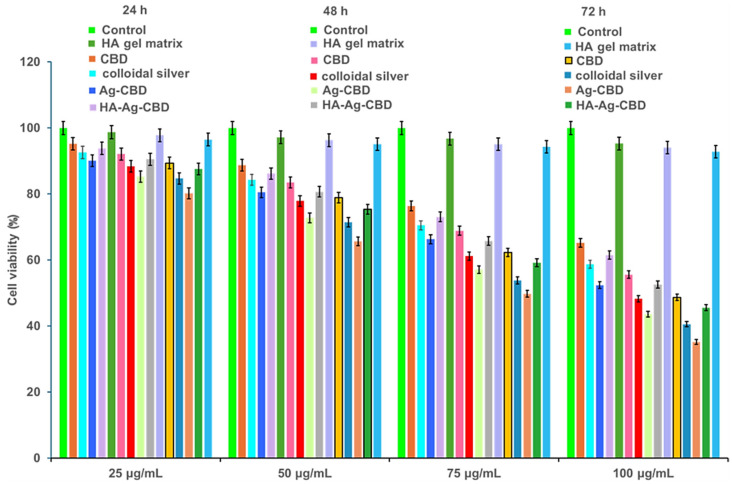
Viability of NHDF cells, assessed at 24, 48, and 72 h after co-incubation with varying concentrations of the HA gel (HA–poloxamer) matrix, CBD, colloidal Ag, the Ag-CBD complex, and the HA-Ag-CBD hybrid gel. Positive control wells included untreated cells, MTT solution, and DMSO. Data are presented as the mean ± SD (n = 3). DMSO: Dimethyl sulfoxide; MTT: 3-(4,5-Dimethylthiazol-2-yl)-2,5-diphenyltetrazolium bromide; NHFD: Normal human dermal fibroblast.

**Figure 12 pharmaceutics-17-01189-f012:**
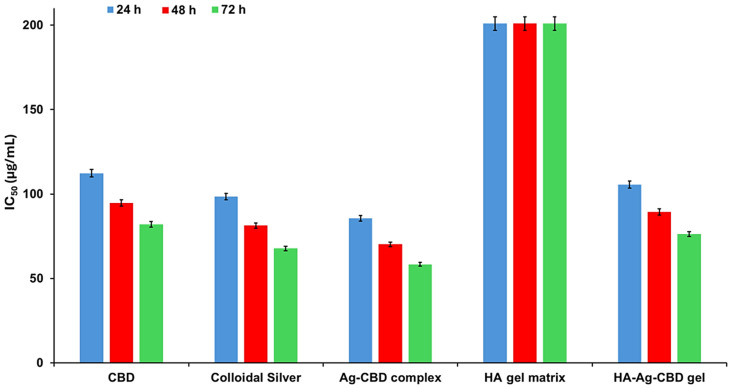
In vitro cytotoxicity of the HA gel (HA–poloxamer) matrix, CBD, colloidal Ag, the Ag-CBD complex, and the HA-Ag-CBD hybrid gel as a function of concentration against NHFD cells after 24, 48, and 72 h. Data are represented as the mean ± SD (n = 3). IC_50_: Half-maximal inhibitory concentration.

**Table 1 pharmaceutics-17-01189-t001:** HA-Ag-CBD hybrid gel composition (per 100 mL of gel).

Component	Concentration	Function
Poloxamer 407	18%	thermoreversible gelation agent
Poloxamer 188	2%	rheology modulator, stabilizer
CBD	0.5%	anti-inflammatory and antioxidant compound
HA	0.1%	moisturizer, film former, promotes tissue repair
AgNPs	0.02%	broad-spectrum antimicrobial agent
Purified water	up to 100 mL	aqueous solvent

AgNPs: Silver nanoparticles; CBD: Cannabidiol; HA: Hyaluronic acid.

**Table 2 pharmaceutics-17-01189-t002:** Evaluation of antibacterial performance against clinically relevant pathogenic strains.

Pathogenic Microorganism	Inhibition Zone Diameter (mm)					
	CBD	Colloidal Ag	Ag-CBD	HA-Ag-CBD	Positive Control (Gentamicin, 100 μg/mL)	Negative Control (HA Gel Matrix, 10 μg/mL)
*Staphylococcus aureus*	9.07 ± 0.35	8.21 ± 0.26	8.64 ± 0.40	9.53 ± 0.30	22.18 ± 0.18	0
*Escherichia coli*	6.81 ± 0.29	6.53 ± 0.44	5.57 ± 0.40	6.05 ± 0.30	20.51 ± 0.28	0
*Pseudomonas aeruginosa*	0	0	0	0	30.48 ± 0.21	0
*Candida albicans*	9.15 ± 0.46	7.18 ± 0.15	7.04 ± 0.50	7.13 ± 0.40	0	0

Values are expressed as the mean ± SD (n = 3). Ag: Silver; SD: Standard deviation.

**Table 3 pharmaceutics-17-01189-t003:** MIC and MBC values of the samples against representative pathogenic strains.

Pathogenic Microorganism	Sample	MIC (μg/mL)	MBC (μg/mL)	Positive Control (Gentamicin, 100 μg/mL)		Negative Control (HA Gel Matrix, 10 μg/mL)	
				MIC (μg/mL)	MBC (μg/mL)	MIC (μg/mL)	MBC (μg/mL)
*Staphylococcus* *aureus*	CBD	1.74 ± 0.25	1.87 ± 0.50	0.63 ± 0.03	0.63 ± 0.04	>400	>400
	colloidal Ag	22.14 ± 0.36	31.05 ± 0.25				
	Ag-CBD	15.37 ± 0.18	20.21 ± 0.18				
	HA-Ag-CBD	15.13 ± 0.41	20.24 ± 0.37				
*Escherichia coli*	CBD	2.89 ± 0.65	2.93 ± 0.37	1.11 ± 0.21	1.11 ± 0.18	>400	>400
	colloidal Ag	10.75 ± 0.32	22.02 ± 0.25				
	Ag-CBD	9.36 ± 0.17	14.21 ± 0.18				
	HA-Ag-CBD	9.24 ± 0.48	387.35 ± 0.28				
*Pseudomonas* *aeruginosa*	CBD	>400	>400	1.93 ± 0.25	1.94± 0.13	>400	>400
	colloidal Ag	>400	>400				
	Ag-CBD	>400	>400				
	HA-Ag-CBD	>400	>400				
*Candida albicans*	CBD	20.72 ± 0.34	22.45 ± 0.41	>400	>400	>400	>400
	colloidal Ag	10.45 ± 0.15	45.22 ± 0.22				
	Ag-CBD	16.35 ± 0.46	34.43 ± 0.54				
	HA-Ag-CBD	16.44 ± 0.27	34.51 ± 0.16				

Values are expressed as the mean ± SD (n = 3). HA gel: HA–poloxamer; MBC: Minimum bactericidal concentration; MIC: Minimum inhibitory concentration.

## Data Availability

The original contributions presented in this study are included in the article. Further inquiries can be directed to the corresponding author.

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
