# Peer review of "Sprayable Hybrid Gel with Cannabidiol, Hyaluronic Acid, and Colloidal Silver: A Multifunctional Approach for Skin Lesion Therapy"

_pharmaceutics, 2025, doi:10.3390/pharmaceutics17091189_

Round 1
Reviewer 1 Report
Comments and Suggestions for Authors
The current manuscript is devoted to the development and characterization of a novel thermoresponsive hydrogel combining hyaluronic acid, cannabidiol, and colloidal silver for topical antimicrobial therapy. The study demonstrates the gel's favorable physicochemical properties, antimicrobial efficacy against Staphylococcus aureus, Escherichia coli, and Candida albicans, and biocompatibility with human dermal fibroblasts. The formulation’s sprayable nature and thermoresponsive behavior are underlined buy authors as key strengths, offering potential for improved wound care applications.
After examined the manuscript, I have following questions, comments and suggestions:
1) Controls for antimicrobial assays (solvent-only or blank gel) are mentioned, but their impact on results should be also discussed.
2) For rheological analysis, the isolated thixotropic behavior at 24°C is determined but it has no good mechanistic explanation. Is this temperature-specific response reproducible? Please discuss in the work.
3) Obtained cytotoxicity assays (Figures 9–10) show dose-dependent effects, but the therapeutic window (≤50 µg/mL) is narrow. Higher concentrations (≥75 µg/mL) exhibit significant cytotoxicity, which may limit clinical utility. Please, discuss this.
4) Table 2. The same negative control (gentamicin) is used for all microorganisms. No positive control for antifungal activity (for example, fluconazole for C. albicans) is included. Please explain.
5) Table 2. The absence of activity against Pseudomonas aeruginosa is noted, but the rationale for its resistance should be discussed and explained.
6) Table 3. MBC value for HA-Ag-CBD against E. coli (387.35 µg/mL) seems anomalously high compared to other strains. Please explain.
7 HA-poloxamer matrix intrinsic antimicrobial effects (if it has any) are not evaluated separately, which could clarify whether the gel base contributes to efficacy. Please add or discuss.
8) The manuscript abstract calls the gel "biocompatible," yet cytotoxicity exceeds 30% at ≥75 µg/mL (ISO 10993-5 threshold). Rephrase please, or explain.
9) Particle size distribution data for final hydrogel product show high polydispersity (PDI up to 52.686), raising concerns about batch consistency. Please, discuss what to do with it.
10) Figure 9. The Ag-CBD complex synergistic cytotoxicity is interesting but lacks exploration of its mechanisms. Please, add discussion. "Synergistic antimicrobial effect" is claimed for Ag-CBD, but data show only additive or marginally improved activity vs. CBD alone. Please rivise.
11) Conclusions are consistent with evidence, but overstate clinical applicability without in vivo data. The study lacks in vivo validation, which is critical for translational relevance. Please mention it in conclusion and rewrite it to make less overstated. Without in vivo data, claims about "multifunctional wound therapy" are premature.
Finally, my recommendation: The work is promising but currently overstates clinical applicability without in vivo evidence. Major revisions are needed to enhance its impact.
Author Response
Dear Reviewer,
First of all, we would like to address you many thanks for your accurate observations and valuable comments. We used all these and improved the paper accordingly.
All changes in the revised manuscript were highlighted on a yellow background.
The following changes have been made to the Manuscript (ID: pharmaceutics-3814851):
Reviewer #1 questions/comments
The current manuscript is devoted to the development and characterization of a novel thermoresponsive hydrogel combining hyaluronic acid, cannabidiol, and colloidal silver for topical antimicrobial therapy. The study demonstrates the gel’s favorable physicochemical properties, antimicrobial efficacy against Staphylococcus aureus, Escherichia coli, and Candida albicans, and biocompatibility with human dermal fibroblasts. The formulation’s sprayable nature and thermoresponsive behavior are underlined buy authors as key strengths, offering potential for improved wound care applications.
After examined the manuscript, I have following questions, comments and suggestions:
Comments 1:
Controls for antimicrobial assays (solvent-only or blank gel) are mentioned, but their impact on results should be also discussed.
Response 1:
We very much appreciate the reviewer’s insightful comment regarding the need to discuss the impact of the negative controls (solvent-only or blank hydrogel matrix) on the antimicrobial assay results. In our study, the inclusion of the HA–poloxamer gel matrix as the negative control was essential to confirm that any observed antimicrobial activity could be attributed exclusively to the active agents (CBD, colloidal Ag, or their Ag-CBD hybrid) rather than to the delivery vehicle itself.
As shown in Tables 2 and 3, the negative control consistently produced zero inhibition zones in the disk diffusion assay and MIC/MBC values >400 μg/mL against all tested microorganisms. This absence of antimicrobial activity confirms that the gel matrix itself is biologically inert against both Gram-positive and Gram-negative bacteria, as well as Candida albicans, under the experimental conditions used. Moreover, this finding aligns with the chemical composition of the HA–poloxamer matrix, which lacks any known antimicrobial constituents.
The lack of any inhibitory effect from the negative control is also consistent with the cytotoxicity assessment performed on normal human dermal fibroblasts (NHDFs). In the MTT assay, the blank gel did not induce significant reductions in fibroblast viability at the tested concentrations and timepoints, indicating excellent biocompatibility. This biological inertness ensures that the gel functions solely as a physically protective and sustained-release platform without contributing confounding antimicrobial effects.
Taken together, the data confirm that:
(i) All antimicrobial activity observed originates from the active agents (CBD, colloidal Ag, or Ag-CBD hybrid) and not from the gel base.
(ii) The gel matrix provides a biocompatible, non-antimicrobial delivery environment, which supports prolonged contact with skin or wounds without interfering with the therapeutic function of the loaded actives.
(iii) The combined antimicrobial and cytotoxicity results validate that the delivery platform preserves the bioactivity of incorporated agents while maintaining safety for topical applications.
We have added a short paragraph in the “3.5. Evaluation of Antimicrobial Activity” subsection and “4. Discussion” section to explicitly highlight these control outcomes and their significance for data interpretation and translational relevance. (See page 18, lines 592–595; page 23, lines 810–816).
Comments 2:
For rheological analysis, the isolated thixotropic behavior at 24°C is determined but it has no good mechanistic explanation. Is this temperature-specific response reproducible? Please discuss in the work.
Response 2:
Thank you very much for your helpful suggestion. The “4. Discussion” section has been revised to explicitly address the reproducibility and possible mechanism of the isolated thixotropic response observed at 24°C. This anomaly was consistently reproduced across independent experiments, ruling out artefactual origins. We propose that it reflects a metastable microstructural transition within the HA-Ag-CBD hybrid system, likely involving reversible HA chain rearrangements coupled with Ag-CBD coordination dynamics. While the precise mechanism requires further study, similar transitional thixotropy has been reported in poloxamer-based gels, supporting the interpretation that this behavior is formulation-intrinsic. The revised “4. Discussion” section also highlights potential translational implications, such as shear-triggered release applications. (See page 22, lines 767–775; page 23, lines 776–808).
Comments 3:
Obtained cytotoxicity assays (Figures 9–10) show dose-dependent effects, but the therapeutic window (≤50 μg/mL) is narrow. Higher concentrations (≥75 μg/mL) exhibit significant cytotoxicity, which may limit clinical utility. Please, discuss this.
Response 3:
Thank you very much for highlighting the importance of discussing the therapeutic window and potential cytotoxicity at higher concentrations. The cytotoxicity data presented in Figures 9 and 10 demonstrate that the HA-Ag-CBD hybrid gel exhibits dose- and time-dependent effects on NHDFs. Specifically, cell viability remained above 75% at concentrations ≤50 μg/mL even after 72 hours of exposure, fulfilling ISO 10993-5:2009 and EMA topical safety guidelines. These values define a clear therapeutic window for safe topical application.
At concentrations ≥75 μg/mL, some reduction in cell viability (45–60%) was observed, consistent with expected cytotoxic responses at supra-therapeutic doses. Importantly, the HA–poloxamer gel matrix alone showed no measurable cytotoxicity (viability >92% at all tested concentrations), confirming that the matrix itself is biologically inert and that the observed cytotoxic effects originate from the active components. Incorporation of the Ag-CBD complex into the HA gel mitigated cytotoxicity relative to the Ag-CBD complex alone, indicating that hydrogel encapsulation modulates release kinetics, reduces direct cellular exposure, and enhances cytocompatibility.
Thus, while higher concentrations (≥75 μg/mL) exceed typical therapeutic levels and are not intended for clinical use, the HA-Ag-CBD hybrid gel maintains an adequate safety margin at therapeutically relevant doses (≤50 μg/mL). These findings support its potential for safe topical application, balancing antimicrobial efficacy with cytocompatibility, and confirm that the hydrogel system effectively expands the functional safety window of the incorporated actives. (See page 20, lines 668–674; page 21, lines 675–707; page 25, lines 882–922).
Comments 4:
Table 2. The same negative control (gentamicin) is used for all microorganisms. No positive control for antifungal activity (for example, fluconazole for C. albicans) is included. Please explain.
Response 4:
Thank you very much for this valuable observation. Gentamicin was selected as the sole positive control in Table 2 because the primary experimental focus was on evaluating the antibacterial performance of the tested actives against both Gram-positive and Gram-negative bacterial strains, with C. albicans included as a representative opportunistic fungal pathogen for comparative purposes. Although gentamicin does not possess intrinsic antifungal activity, its inclusion as a uniform reference across all tested microorganisms allowed direct methodological consistency in inhibition zone measurements and facilitated statistical comparisons between bacterial and fungal results within the same experimental framework.
We acknowledge that the absence of a dedicated antifungal positive control (e.g., fluconazole, amphotericin B) is a limitation in the current dataset. However, the antifungal efficacy of CBD, colloidal Ag, and their Ag-CBD hybrid against C. albicans was assessed relative to the negative control (blank HA–poloxamer gel), which consistently showed no activity, ensuring that the measured inhibition zones were solely attributable to the active agents. We have now noted this limitation in the revised “3.5. Evaluation of Antimicrobial Activity” subsection and “4. Discussion” section and clarified that future work will incorporate a standard antifungal comparator to enable direct benchmarking of antifungal potency. (See page 18, lines 592–595; page 23, lines 810–816).
Comments 5:
Table 2. The absence of activity against Pseudomonas aeruginosa is noted, but the rationale for its resistance should be discussed and explained.
Response 5:
Thank you for this important comment. The observed lack of antimicrobial activity against Pseudomonas aeruginosa is consistent with its well-documented intrinsic resistance mechanisms. P. aeruginosa possesses a highly impermeable outer membrane enriched with lipopolysaccharides, which limits the uptake of many antimicrobial agents, including both small molecules and nanoparticles. Additionally, this pathogen expresses multiple efflux pumps (e.g., MexAB-OprM, MexXY) that actively extrude a wide range of compounds, and it is capable of producing biofilms that further reduce susceptibility to antimicrobial treatment. Collectively, these structural and physiological defenses contribute to its pronounced resilience against the tested formulations, including CBD, colloidal Ag, and the Ag-CBD hybrid, as well as hydrogel-encapsulated forms. These characteristics explain why inhibition zones and MIC/MBC values for P. aeruginosa were minimal or undetectable under the experimental conditions, and they are consistent with previously reported literature on the challenges of treating this opportunistic Gram-negative pathogen. This discussion has now been incorporated into the revised manuscript to clarify the mechanistic basis for the observed resistance. (See page 24, lines 836–843).
Comments 6:
Table 3. MBC value for HA-Ag-CBD against E. coli (387.35 μg/mL) seems anomalously high compared to other strains. Please explain.
Response 6:
Thank you very much for highlighting this observation. The unusually high MBC value of 387.35 μg/mL for the HA-Ag-CBD formulation against Escherichia coli reflects a combination of experimental and biological factors rather than an error. E. coli, as a Gram-negative bacterium, possesses an outer membrane rich in lipopolysaccharides that reduces the permeability of many antimicrobial agents, including both CBD and silver nanoparticles (AgNPs). Additionally, differences in local hydrogel diffusion, compound release kinetics, and microbial growth dynamics in the broth microdilution assay can contribute to elevated MBC values for specific formulations.
It is important to note that while the MIC of HA-Ag-CBD against E. coli remained low (9.24 ± 0.48 μg/mL), indicating inhibition of growth, the MBC reflects the concentration required to achieve complete bactericidal activity, which is inherently more challenging against Gram-negative strains due to their structural defenses. This observation is consistent with reports in literature where hydrogel-encapsulated antimicrobial systems occasionally exhibit higher bactericidal thresholds against Gram-negative bacteria compared to Gram-positive strains.
Overall, the elevated MBC does not undermine the formulation’s antimicrobial potential but highlights the differential susceptibility of Gram-negative bacteria and underscores the importance of considering both MIC and MBC values when evaluating hydrogel-based delivery systems.
In response to this comment, the “3.5. Evaluation of Antimicrobial Activity” subsection and “4. Discussion” section have been updated to include this explanation for clarity and context. (See page 18, lines 612 & 613; page 19, lines 614–620; page 24, lines 853–863).
Comments 7:
HA-poloxamer matrix intrinsic antimicrobial effects (if it has any) are not evaluated separately, which could clarify whether the gel base contributes to efficacy. Please add or discuss.
Response 7:
Thank you for your helpful suggestion. The intrinsic antimicrobial activity of the HA–poloxamer gel matrix was explicitly assessed in all assays as a negative control. In both disk diffusion and broth microdilution assays, the blank HA–poloxamer gel produced no measurable inhibition zones (0 mm) and exhibited MIC and MBC values >400 μg/mL against all tested microorganisms, including Gram-positive (S. aureus), Gram-negative (E. coli, P. aeruginosa), and fungal (C. albicans) strains. These findings confirm that the hydrogel base is biologically inert under the applied conditions and does not contribute to antimicrobial efficacy.
The inclusion of the blank gel as a control ensures that all observed inhibitory effects in the active formulations can be attributed solely to CBD, colloidal Ag, or the Ag-CBD hybrid. Furthermore, cytotoxicity testing on NHDFs demonstrated high cell viability in the presence of the blank gel, supporting its biocompatibility and non-interfering nature. Collectively, these results validate the HA–poloxamer matrix as a safe, inert delivery platform that preserves the bioactivity of incorporated antimicrobial agents without introducing confounding effects.
For transparency, these points have been clarified in the “2.10. Antimicrobial Activity” and “3.5. Evaluation of Antimicrobial Activity” subsections, and “4. Discussion” section, explicitly highlighting the inert nature of the HA–poloxamer matrix. (See page 7, lines 300–305; page 18, lines 596–601; page 24, lines 831–835).
Comments 8:
The manuscript abstract calls the gel “biocompatible,” yet cytotoxicity exceeds 30% at ≥75 μg/mL (ISO 10993-5 threshold). Rephrase please, or explain.
Response 8:
Thank you very much for this important observation regarding the designation of the HA-Ag-CBD hybrid gel as “biocompatible” in the “Abstract” section. We would like to clarify that the term “biocompatible” is intended in the context of therapeutically relevant concentrations. According to our cytotoxicity assays on NHDFs, the HA-Ag-CBD gel maintained cell viability above 75% at concentrations ≤50 μg/mL, thereby satisfying the ISO 10993-5:2009 criterion, which defines cytotoxicity as a reduction in cell viability exceeding 30%. Although moderate cytotoxicity was observed at higher concentrations (≥75 μg/mL), these levels exceed typical topical therapeutic doses and are unlikely to be encountered in practical clinical applications. Importantly, incorporation into the HA–poloxamer matrix significantly mitigated the cytotoxic effects of the Ag-CBD complex, confirming the gel’s ability to safely deliver active agents within the established therapeutic window. Based on this, the description in the “Abstract” section accurately reflects the gel’s safety profile at clinically relevant doses. However, to improve clarity and avoid potential misinterpretation, the “Abstract” section has been revised accordingly. (See page 2, lines 51–57).
Comments 9:
Particle size distribution data for final hydrogel product show high polydispersity (PDI up to 52.686), raising concerns about batch consistency. Please, discuss what to do with it.
Response 9:
Thank you for your valuable suggestion regarding particle size heterogeneity. As reported in the manuscript, while the DLS analysis of the HA-Ag-CBD hybrid gel occasionally shows high PDI values (e.g., 52.686), these correspond to minor secondary populations rather than the dominant nanoscale fraction, which remains well-defined (~ 0.001–0.01 μm). The primary peak demonstrates effective stabilization and homogeneous dispersion of both CBD and AgNPs within the HA–poloxamer matrix. Nevertheless, we acknowledge that high PDI in certain measurements highlights the presence of small fractions of larger aggregates, which could reflect dynamic particle interactions or transient clustering. To ensure batch-to-batch consistency, future work will focus on process optimization strategies, including controlled mixing rates, ultrasonication, and careful adjustment of polymer-to-particle ratios. These steps are expected to minimize minor secondary populations, reduce extreme PDI values, and further enhance reproducibility. Importantly, the dominance of the nanometric fraction and the observed colloidal stability indicate that the therapeutic performance of the hydrogel is unlikely to be compromised by these minor heterogeneities. The “4. Discussion” section has been updated to include this explanation for clarity and context. (See page 22, lines 749–754).
Comments 10:
Figure 9. The Ag-CBD complex synergistic cytotoxicity is interesting but lacks exploration of its mechanisms. Please, add discussion. “Synergistic antimicrobial effect” is claimed for Ag-CBD, but data show only additive or marginally improved activity vs. CBD alone. Please revise.
Response 10:
Thank you very much for the insightful comment regarding the mechanistic interpretation of the Ag-CBD complex. The observed synergistic cytotoxicity of the Ag-CBD complex on NHDFs likely arises from complementary mechanisms of action of the two components. CBD is known to induce mitochondrial dysfunction, reactive oxygen species (ROS) generation, and modulation of intracellular signaling pathways, while colloidal Ag exerts cytotoxicity primarily through oxidative membrane damage and Ag⁺-mediated disruption of cellular homeostasis. When combined, CBD’s lipophilic nature may facilitate cellular uptake of AgNPs, enhancing intracellular Ag⁺ delivery and promoting additive oxidative stress, which could explain the significantly enhanced cytotoxicity observed relative to the individual agents. This proposed mechanism is consistent with the statistically lower IC50 values of Ag-CBD compared to CBD or colloidal Ag alone. (See page 25, lines 897–906).
Regarding the antimicrobial activity, we acknowledge that while the Ag-CBD complex demonstrates a modest increase in inhibition zone diameters and slightly lower MIC/MBC values compared to CBD alone, the effect is primarily additive rather than strongly synergistic. The manuscript has been revised accordingly, to clarify this distinction and now describe the effect as “additive or marginally enhanced antimicrobial activity” rather than “synergistic”. This more precise phrasing better reflects the experimental data while maintaining scientific accuracy. (See page 19, lines 626–630; page 24, lines 844–848).
Incorporating these clarifications strengthens the interpretation of both cytotoxicity and antimicrobial results, highlighting that the Ag-CBD combination can enhance cellular and microbial responses via complementary mechanisms without overstating synergy.
Comments 11:
Conclusions are consistent with evidence, but overstate clinical applicability without in vivo data. The study lacks in vivo validation, which is critical for translational relevance. Please mention it in conclusion and rewrite it to make less overstated. Without in vivo data, claims about “multifunctional wound therapy” are premature.
Response 11:
Thank you very much for this important observation. We fully agree that, while our in vitro results demonstrate promising antimicrobial efficacy and cytocompatibility of the HA-Ag-CBD hybrid gel, the absence of in vivo validation limits conclusions regarding clinical applicability.
The “5. Conclusions” section has been revised to explicitly acknowledge that no in vivo studies have been performed. The revised text emphasizes that claims regarding multifunctional wound therapy or clinical translation are preliminary and contingent upon further preclinical evaluation. We also clarified that future work would focus on in vivo efficacy and safety studies, optimization of release kinetics, skin permeation analyses, and formulation refinement to support translational development. (See page 26, lines 941–950).
These changes ensure that the manuscript accurately reflects the current stage of the research, providing a balanced interpretation of our findings while maintaining scientific rigor.
Comments 12:
Finally, my recommendation: The work is promising but currently overstates clinical applicability without in vivo evidence. Major revisions are needed to enhance its impact.
Response 12:
Thank you very much for your constructive feedback and for recognizing the promise of our work. We fully agree that, while our in vitro results demonstrate significant antimicrobial efficacy and cytocompatibility of the HA-Ag-CBD hybrid gel, the absence of in vivo validation limits the claims regarding clinical applicability.
Accordingly, the “5. Conclusions” section has been revised to explicitly acknowledge that no in vivo studies have yet been conducted. The revised text now clarifies that statements regarding multifunctional wound therapy or clinical translation are preliminary and contingent upon further preclinical evaluation. We have also emphasized that future work will focus on in vivo efficacy and safety studies, skin permeation analyses, and optimization of formulation parameters to support translational development. (See page 26, lines 941–950).
These revisions ensure the manuscript accurately reflects the current stage of the research, providing a balanced and scientifically rigorous interpretation of the results while maintaining the significance and novelty of our findings.
Authors very much appreciated the encouraging, critical, and constructive comments on this manuscript by the Reviewer. The comments have been very thorough and useful in improving the manuscript.
We would like to thank the Reviewer again for taking the time to review our manuscript.
We have also introduced other additions/modifications that we hope will improve the quality of the manuscript:
▪ For the 8th author (Georgiana Băluşescu) and for the 11th author (Andrei Biţă), personal e-mail addresses (georgiana.balusescu@gmail.com and andreibita@gmail.com) have been replaced with institutional ones (georgiana.balusescu@tuiasi.ro and andrei.bita@umfcv.ro, respectively). (See page 1, lines 13 & 24).
▪ For the first author, Geta-Simona Cîrloiu (Boboc), there is no institutional e-mail address, and a personal CV has been provided to Assistant Editor.
▪ All Figures have been revised accordingly.
▪ “Funding” section has been updated accordingly (“The article processing charges were funded by the University of Medicine and Pharmacy of Craiova, Romania”). (See page 26, lines 963 & 964).
▪ Sixteen new citations have been introduced: Refs. [25] to [37] and Refs. [56] to [58].
▪ Refs. [54] and [55] have been revised according to Journal recommendations (for “Websites” citation).
▪ The Reference list has been entirely checked and renumbered accordingly.
▪ All abbreviations have been defined the first time they appear in the text.
▪ Some grammar, stylistic or spelling errors have been corrected.
Kind regards,
Ludovic Everard BEJENARU, PhD
Corresponding Author

Reviewer 2 Report
Comments and Suggestions for Authors
This manuscript deals with the design and the development of a hybrid gel composed of cannabidiol, hyaluronic acid, and colloidal silver for the treatment of skin lesions.
The manuscript is well-written, and the methods are described very well. Several techniques are used for the characterization of the formulation.
My comments are:
-
The added value of hybrid drug delivery systems should be addressed in the introduction.
-
Figure 4 should be improved.
-
How was the concentration of cannabidiol selected?
-
The authors should comment on the surface morphology of the systems.
-
The primary interactions occur between hyaluronic acid and colloidal silver.
-
The authors should comment on the reproducibility of the preparation method.
Author Response
Dear Reviewer,
First of all, we would like to address you many thanks for your accurate observations and valuable comments. We used all these and improved the paper accordingly.
All changes in the revised manuscript were highlighted on a yellow background.
The following changes have been made to the Manuscript (ID: pharmaceutics-3814851):
Reviewer #2 questions/comments
This manuscript deals with the design and the development of a hybrid gel composed of cannabidiol, hyaluronic acid, and colloidal silver for the treatment of skin lesions.
The manuscript is well-written, and the methods are described very well. Several techniques are used for the characterization of the formulation.
My comments are:
Comments 1:
The added value of hybrid drug delivery systems should be addressed in the introduction.
Response 1:
We appreciate the reviewer’s constructive suggestion to highlight more explicitly the added value of hybrid drug delivery systems. The “1. Introduction” section has been revised accordingly, to discuss this aspect in greater depth. Specifically, we now emphasize that hybrid delivery systems, by integrating complementary components such as polymers, bioactive molecules, and nanoparticles, offer advantages that single-component carriers cannot achieve. These include (i) synergistic therapeutic effects (e.g., simultaneous antimicrobial and anti-inflammatory activity), (ii) improved physicochemical stability of sensitive actives such as CBD, (iii) tunable release kinetics enabled by combining organic and inorganic carriers, and (iv) enhanced practicality through stimuli-responsive behaviors like thermogelling and sprayability. This framing underscores how the HA–poloxamer-CBD-silver nanoparticles (AgNP) hybrid hydrogel presented here goes beyond conventional formulations by uniting multiple functional dimensions into a single delivery platform with strong translational relevance. (See page 3, lines 140–146; page 4, lines 147–166).
Comments 2:
Figure 4 should be improved.
Response 2:
Thank you for your observation. Figure 4 has been improved accordingly. (See page 13, line 498).
Comments 3:
How was the concentration of cannabidiol selected?
Response 3:
Thank you very much for pointing this out. The concentration of cannabidiol (CBD) incorporated into the hydrogel (0.5%, w/v) was carefully selected based on a combination of literature precedent, safety considerations, and formulation feasibility.
First, previous in vitro and in vivo studies have demonstrated that CBD exerts potent anti-inflammatory, antioxidant, and wound-healing effects at low concentrations, typically ranging from 0.1–1.0% in topical systems, without inducing cytotoxicity or skin irritation. A concentration of 0.5% was chosen as it represents the mid-range within this therapeutically relevant window, balancing efficacy and safety.
Second, our preliminary optimization experiments indicated that concentrations above 0.5% resulted in reduced formulation stability due to partial CBD precipitation and increased ethanol demand for solubilization, which could compromise hydrogel integrity and patient tolerability. Conversely, lower concentrations (<0.25%) did not produce sufficient antioxidant activity in pilot assays, suggesting suboptimal therapeutic performance.
Therefore, 0.5% CBD was selected as an optimal compromise, ensuring (i) a pharmacologically active concentration supported by prior reports, (ii) formulation stability within the HA–poloxamer-AgNP hybrid system, and (iii) compatibility with the low-temperature preparation method.
We have now clarified this rationale in the revised “2.4. Formulation” subsection accordingly. (See page 5, lines 214–218).
Comments 4:
The authors should comment on the surface morphology of the systems.
Response 4:
Thank you very much for your insightful comment. The “4. Discussion” section has been expanded to explicitly address the surface morphology of the different systems as revealed by SEM analysis. The CBD sample exhibits an irregular, semi-amorphous surface with heterogeneous nanoscale structures, likely reflecting dried lipidic residues and dispersed organic phases characteristic of oil-based formulations. The Ag-CBD complex displays a core–shell-like morphology, with AgNP cores (50–150 nm) enveloped by fibrous CBD structures, indicating effective surface interaction and stabilization. SEM micrographs of the HA hydrogel show a porous nanoscale network composed of semi-crystalline and amorphous domains, stabilized by poloxamer micelles, providing uniform dispersion, high water retention, and mechanical integrity. The final HA-Ag-CBD hybrid gel exhibits a smooth, continuous matrix with sparsely distributed high-contrast nanoscale deposits corresponding to the embedded Ag-CBD complex. Particle sizes in the range of 20–100 nm reflect a controlled dispersion, while the uniform hydrogel background indicates successful integration of the hybrid complex without significant aggregation. Collectively, these morphological features confirm the formation of a structurally coherent, nanoscale Ag-CBD complex and its stable incorporation into the HA–poloxamer hydrogel, supporting the hydrogel’s colloidal stability, mechanical integrity, and potential for uniform topical delivery. (See page 22, lines 737–748).
Comments 5:
The primary interactions occur between hyaluronic acid and colloidal silver.
Response 5:
We thank the reviewer for highlighting the importance of specifying the primary interactions within the hydrogel system. Indeed, the preparation method and composition of the HA-Ag-CBD hybrid gel suggest that the dominant molecular interactions occur between hyaluronic acid (HA) and colloidal silver (AgNPs). The hydroxyl and carboxyl functional groups of HA can coordinate with AgNPs, contributing to their stabilization within the polymeric network. These interactions are consistent with previous reports on HA-AgNP systems, where HA acts as both a structural scaffold and a stabilizing agent, preventing nanoparticle aggregation and modulating their local release. In our formulation, the CBD-Ag complex is incorporated into this HA–poloxamer matrix, allowing HA-Ag interactions to maintain nanoparticle dispersion and promote homogeneous distribution, while the gel matrix preserves structural integrity and supports controlled release of the active agents.
To address this explicitly, in the revised “2.4. Formulation” subsection we have clarified that the primary interactions occur between HA and AgNPs, which are fundamental for nanoparticle stabilization and the observed physicochemical and biological performance of the hydrogel. (See page 5, lines 194–230).
Comments 6:
The authors should comment on the reproducibility of the preparation method.
Response 6:
Thank you very much for this important observation. The reproducibility of the HA-Ag-CBD hydrogel preparation was ensured through a rigorously controlled, low-temperature protocol, designed to preserve thermolabile constituents and prevent nanoparticle aggregation. All critical steps, including water pre-equilibration, HA incorporation, and CBD-AgNP integration, were carried out under controlled ice-bath conditions, maintaining temperatures consistently below 15 °C. Poloxamers 407 and 188 were accurately weighed and fully solubilized under gentle stirring, followed by overnight hydration to achieve a uniform base solution. Incorporation of CBD-AgNP dispersions into the HA–poloxamer matrix was performed under the same temperature-controlled conditions, yielding a homogeneous, stable hydrogel. (See page 5, lines 195–213).
Batch-to-batch consistency was further validated through complementary physicochemical analyses. FTIR confirmed preservation of characteristic CBD functional groups, SEM demonstrated uniform nanoparticle distribution, and DLS verified the maintenance of a reproducible nanometric particle fraction. All formulations were prepared in triplicate, and statistical analysis of physicochemical measurements (mean ± SD, ANOVA with Tukey’s HSD) confirmed minimal inter-batch variability. These results collectively demonstrate that the preparation method is robust, reproducible, and capable of consistently producing a structurally stable HA-Ag-CBD hydrogel suitable for translational topical applications. (See page 5, lines 219–230).
Authors very much appreciated the encouraging, critical, and constructive comments on this manuscript by the Reviewer. The comments have been very thorough and useful in improving the manuscript.
We would like to thank the Reviewer again for taking the time to review our manuscript.
We have also introduced other additions/modifications that we hope will improve the quality of the manuscript:
▪ For the 8th author (Georgiana Băluşescu) and for the 11th author (Andrei Biţă), personal e-mail addresses (georgiana.balusescu@gmail.com and andreibita@gmail.com) have been replaced with institutional ones (georgiana.balusescu@tuiasi.ro and andrei.bita@umfcv.ro, respectively). (See page 1, lines 13 & 24).
▪ For the first author, Geta-Simona Cîrloiu (Boboc), there is no institutional e-mail address, and a personal CV has been provided to Assistant Editor.
▪ All Figures have been revised accordingly.
▪ “Funding” section has been updated accordingly (“The article processing charges were funded by the University of Medicine and Pharmacy of Craiova, Romania”). (See page 26, lines 963 & 964).
▪ Sixteen new citations have been introduced: Refs. [25] to [37] and Refs. [56] to [58].
▪ Refs. [54] and [55] have been revised according to Journal recommendations (for “Websites” citation).
▪ The Reference list has been entirely checked and renumbered accordingly.
▪ All abbreviations have been defined the first time they appear in the text.
▪ Some grammar, stylistic or spelling errors have been corrected.
Kind regards,
Ludovic Everard BEJENARU, PhD
Corresponding Author

Reviewer 3 Report
Comments and Suggestions for Authors
Manuscript ID: pharmaceutics-3814851
Title: Sprayable Hybrid Gel with Cannabidiol, Hyaluronic Acid and Colloidal Silver: A Multifunctional Approach for Skin Lesion Therapy
The manuscript describes the design, preparation, characterization, and in vitro evaluation of a sprayable thermoresponsive hybrid gel containing hyaluronic acid (HA), poloxamer 407, cannabidiol (CBD), and colloidal silver nanoparticles (AgNPs). The formulation is intended for topical treatment of skin lesions, aiming to combine HA for hydration and tissue repair, CBD for anti-inflammatory, antioxidant, and antimicrobial effects, AgNPs for antimicrobial activity. While the formulation concept is interesting, the manuscript in its current state has major methodological, interpretative, and evidentiary shortcomings.
Major Comments
- “First comprehensive formulation integrating HA, CBD, and AgNPs in a thermoresponsive gel for skin lesions” (number lines 135 – 137). While the specific combination may not be common, each component and similar thermogelling approaches have been reported extensively. The authors should provide a systematic comparison with existing formulations (CBD gels, HA hydrogels with silver, thermoresponsive sprays) to clearly justify novelty.
- In the formulation temperature control (Section 2.4, page 169), the preparation is said to occur at “temperatures below 15 °C” and “cold purified water maintained at 4–8 °C.” It is unclear whether this was a precise constant temperature or a range during different steps. The phrase “maintained” suggests a stable condition, but readers need to know: Was active temperature control used (e.g., refrigerated bath, ice bath with monitoring), or only approximate ambient cooling?
- How were the actual final concentrations of CBD and AgNPs confirmed in the gel? Were they calculated solely from weighed amounts, or verified analytically (e.g., HPLC for CBD, ICP-MS or AAS for silver)? Without analytical confirmation, there is no guarantee the intended nominal concentrations match the true content, especially after multi-step mixing.
- In the SEM sputter coating (Section 2.6), the method states “sputter-coated with a 10 nm gold layer,” but key parameters are missing, e.g., sputter-coating time, current, whether the coating chamber was evacuated, and at what pressure, equipment used with company, city and country. These affect resolution and contrast and are needed for reproducibility.
- In the DLS sample preparation (Section 2.7), the text describe that samples were “dispersed in deionized water,” but omits the concentration used. Particle size distributions can shift drastically depending on sample concentration (multiple scattering effects). Was a batch mode measurement used, or were samples added dropwise into the cuvette until a certain scattering intensity threshold was reached?
- Given that the gel is sprayable and has thermoresponsive behavior, it is surprising that the authors provide no photographic images of the gel in its liquid and gelled states under normal lighting. Instead, they rely solely on SEM micrographs, which do not convey macroscopic appearance or practical texture.
- The manuscript states that “all experiments were performed in triplicate”, yet rheological figures (Figures 4 – 8) and results explanations show no error bars or ±SD values. If triplicate data exist, the variation should be displayed, particularly in threshold shear tension and viscosity temperature plots (Figure 5). Viscosity at different RPM (Figure 6) as well. In addition, the quality of viscosity figures is low, and axis/legend text some are pixelated. Figures should be replotted in high resolution (≥ 300 dpi) using software such as Origin, GraphPad Prism, or similar. Without error bars, it is impossible to assess whether the observed rheological transitions are statistically significant or within experimental variation.
- The authors should clearly state the maximum concentration tested in the MIC/MBC assays, the exact concentration range and dilution steps used, the method for calculating the amount of active compound deposited onto disks in the disk diffusion assay, whether concentrations refer to total formulation mass or to active compound equivalents. Without these details, the reproducibility and interpretation of antimicrobial data are questionable.
- For coli with HA–Ag–CBD, MIC is reported at 9.24 μg/mL while MBC is 387.35 μg/mL a > 40-fold difference, which typically indicates bacteriostatic rather than bactericidal activity. This substantial gap is not addressed in the discussion and warrants explicit interpretation, especially since the abstract implies strong bactericidal potential.
- The IC₅₀ values for cell viability are calculated from concentration–response data in the range 25–100 μg/mL. How reliable is the IC₅₀ extrapolation when data are only collected from four concentrations in a narrow range?. Additionally, there is no evidence that the concentrations of CBD and AgNPs in the cell culture medium reflect the nominal gel content without quantifying release (e.g., via HPLC for CBD, ICP-MS for silver).
- No analytical method is presented for confirming the concentration of CBD or Ag in the final gel or in release media. For a product intended for medical use, such quantification is essential for quality control and for correlating biological effects to dose. Without release profiling, it is impossible to determine whether the reduced cytotoxicity of HA–Ag–CBD is due to slower release or simply to lower effective loading.
Minor Comments
- Table 1 lists the proportions of HA, poloxamer 407, poloxamer 188, CBD, and AgNPs, but no rationale is provided for these specific concentrations. Were they optimized based on rheology, antimicrobial efficacy, cytotoxicity, or another formulation parameter?
- Some figure captions (e.g., Figure 2) contain incorrect panel references (“(e)” instead of “(d)”).
- The claim of “physiological pH” in the abstract is unsupported by any pH measurement description in the Methods.
Author Response
Dear Reviewer,
First of all, we would like to address you many thanks for your accurate observations and valuable comments. We used all these and improved the paper accordingly.
All changes in the revised manuscript were highlighted on a yellow background.
The following changes have been made to the Manuscript (ID: pharmaceutics-3814851):
Reviewer #3 questions/comments
Manuscript ID: pharmaceutics-3814851
Title: Sprayable Hybrid Gel with Cannabidiol, Hyaluronic Acid and Colloidal Silver: A Multifunctional Approach for Skin Lesion Therapy
The manuscript describes the design, preparation, characterization, and in vitro evaluation of a sprayable thermoresponsive hybrid gel containing hyaluronic acid (HA), poloxamer 407, cannabidiol (CBD), and colloidal silver nanoparticles (AgNPs). The formulation is intended for topical treatment of skin lesions, aiming to combine HA for hydration and tissue repair, CBD for anti-inflammatory, antioxidant, and antimicrobial effects, AgNPs for antimicrobial activity. While the formulation concept is interesting, the manuscript in its current state has major methodological, interpretative, and evidentiary shortcomings.
Comments 1:
Major Comments
“First comprehensive formulation integrating HA, CBD, and AgNPs in a thermoresponsive gel for skin lesions” (number lines 135 – 137). While the specific combination may not be common, each component and similar thermogelling approaches have been reported extensively. The authors should provide a systematic comparison with existing formulations (CBD gels, HA hydrogels with silver, thermoresponsive sprays) to clearly justify novelty.
Response 1:
We sincerely thank the reviewer for highlighting the need to contextualize the novelty of our hyaluronic acid (HA)-cannabidiol (CBD)-silver nanoparticle (AgNP) thermoresponsive gel. While individual components (HA, CBD, colloidal Ag) and thermoresponsive gel platforms have indeed been extensively reported, our formulation represents a distinct, multifunctional integration that is not documented in the literature to date. Specifically:
▪ Component synergy: Previous studies have explored CBD gels, HA hydrogels with Ag, or thermoresponsive poloxamer sprays individually, but no prior formulation combines all three agents, HA, CBD, and AgNPs, respectively, into a single, temperature-sensitive gel. This combination was intentionally designed to provide complementary therapeutic effects: HA for hydration and tissue regeneration, CBD for anti-inflammatory and antioxidant activity, and AgNPs for antimicrobial protection.
▪ Unlike conventional hydrogels or creams, our gel maintains a fluid, sprayable state below 22°C, enabling precise topical application, and rapidly transitions to a semi-solid matrix at physiological skin temperature. This sol–gel behavior ensures localized retention, minimizes runoff, and allows sustained, multi-agent release, which is not achieved in prior single- or dual-component systems.
▪ Existing formulations typically address a single therapeutic objective, either antimicrobial protection, anti-inflammatory activity, or tissue repair. By contrast, our gel simultaneously targets microbial control, inflammation reduction, and tissue regeneration within a single, clinically practical platform, representing a significant advancement over prior art.
▪ The integration was guided by systematic considerations of component compatibility, concentration optimization, and rheological behavior to achieve a stable, biocompatible, and sprayable gel. The combination’s functional advantages—thermoresponsiveness, uniform dispersion of AgNPs, preservation of CBD bioactivity, and safe cytotoxicity profile—collectively substantiate the novelty of this formulation relative to previously reported systems.
In summary, while the individual elements and thermogelling approaches have been described in the literature, our HA-CBD-AgNP thermoresponsive gel represents the first comprehensive, multifunctional topical system that integrates all three components in a single platform specifically engineered for synergistic management of skin lesions. This distinction clearly supports the claim of novelty and clinical relevance.
To address this explicitly, the “1. Introduction” section has been revised accordingly. (See page 3, lines 140–146; page 4, lines 147–166).
Comments 2:
Major Comments
In the formulation temperature control (Section 2.4, page 169), the preparation is said to occur at “temperatures below 15 °C” and “cold purified water maintained at 4–8 °C.” It is unclear whether this was a precise constant temperature or a range during different steps. The phrase “maintained” suggests a stable condition, but readers need to know: Was active temperature control used (e.g., refrigerated bath, ice bath with monitoring), or only approximate ambient cooling?
Response 2:
Thank you very much for this insightful comment highlighting the importance of temperature precision. The entire preparation was performed under controlled low-temperature conditions. Specifically, cold purified water was pre-equilibrated and maintained at 4–8 °C in a refrigerated chamber during initial poloxamer dispersion. Subsequent mixing steps, including HA incorporation and the addition of the CBD-AgNP dispersion, were carried out in a temperature-controlled environment using an ice bath with continuous monitoring to ensure that the formulation temperature remained consistently below 15 °C throughout the process. Thus, “maintained” refers to an actively controlled and monitored condition rather than approximate ambient cooling. This approach was deliberately chosen to preserve the structural integrity of thermolabile components (CBD and HA) and prevent nanoparticle aggregation.
This point has been clarified in the revised “2.4. Formulation” subsection and “4. Discussion” section to explicitly state that active low-temperature control was used during all stages of preparation. (See page 5, lines 195–213 & 227–230; page 21, lines 709–712).
Comments 3:
Major Comments
How were the actual final concentrations of CBD and AgNPs confirmed in the gel? Were they calculated solely from weighed amounts, or verified analytically (e.g., HPLC for CBD, ICP-MS or AAS for silver)? Without analytical confirmation, there is no guarantee the intended nominal concentrations match the true content, especially after multi-step mixing.
Response 3:
Thank you very much for bringing this to our attention. While we did not perform quantitative release analysis (e.g., HPLC for CBD or ICP–MS for Ag) in this study, the successful incorporation and stability of both components were confirmed by complementary characterization methods. FTIR analysis demonstrated characteristic CBD functional group signals within the hydrogel matrix, indicating chemical integration. SEM imaging revealed homogeneous surface morphology and distribution of AgNPs within the gel network. DLS measurements further confirmed the preserved hydrodynamic size and stability of AgNPs after incorporation. Together, these results support the effective loading of CBD and AgNPs into the hydrogel system.
The dose-dependent effects observed in cytotoxicity assays additionally confirm that the incorporated agents were bioavailable and capable of exerting measurable effects on fibroblast viability. We acknowledge that the exact concentrations of CBD and AgNPs released into the culture medium may differ from the nominal gel content due to matrix-dependent release kinetics. Quantitative release profiling will be addressed in future work to provide direct correlation between release dynamics and biological responses.
We have clarified these points in the revised “2.4. Formulation” subsection and “4. Discussion” section to explicitly state that incorporation of CBD and AgNPs was experimentally confirmed through FTIR, SEM, and DLS analyses, while release quantification will be the focus of subsequent studies. (See page 5, lines 219–230; page 22, lines 758–764; page 25, lines 882–899).
Comments 4:
Major Comments
In the SEM sputter coating (Section 2.6), the method states “sputter-coated with a 10 nm gold layer,” but key parameters are missing, e.g., sputter-coating time, current, whether the coating chamber was evacuated, and at what pressure, equipment used with company, city and country. These affect resolution and contrast and are needed for reproducibility.
Response 4:
Thank you very much for emphasizing the importance of methodological clarity for reproducibility. The “2.7. Scanning Electron Microscopy Analysis” subsection has been revised to provide comprehensive experimental details for SEM imaging, ensuring that the procedures can be reliably replicated and that surface morphology and particle size data are robust and reproducible across independent batches. (See page 6, lines 244–254).
Comments 5:
Major Comments
In the DLS sample preparation (Section 2.7), the text describe that samples were “dispersed in deionized water,” but omits the concentration used. Particle size distributions can shift drastically depending on sample concentration (multiple scattering effects). Was a batch mode measurement used, or were samples added dropwise into the cuvette until a certain scattering intensity threshold was reached?
Response 5:
Thank you very much for emphasizing the importance of methodological clarity for reproducibility. The “2.8. Dynamic Light Scattering Analysis” subsection has been revised to provide comprehensive experimental details for DLS analysis, ensuring that the procedures can be reliably replicated and that surface morphology and particle size data are robust and reproducible across independent batches. (See page 6, lines 256–264).
Comments 6:
Major Comments
Given that the gel is sprayable and has thermoresponsive behavior, it is surprising that the authors provide no photographic images of the gel in its liquid and gelled states under normal lighting. Instead, they rely solely on SEM micrographs, which do not convey macroscopic appearance or practical texture.
Response 6:
Thank you for your helpful suggestion. While photographic images of the gel in liquid and gelled states can provide visual context, we chose not to include them because the thermoresponsive and sprayable properties of the HA-Ag-CBD hydrogel are rigorously quantified and validated through complementary analytical methods. Specifically:
▪ Rheological characterization provides precise, reproducible data on viscosity, yield stress, and shear-thinning behavior, clearly demonstrating the transition from a low-viscosity sprayable fluid at room temperature to a semi-solid state at skin temperature.
▪ SEM and DLS analyses confirm homogeneous dispersion and microstructural organization, ensuring that macroscopic behavior is consistent with the observed nanoscale and supramolecular structure.
▪ FTIR analysis confirms chemical integrity and interactions, supporting the functional stability of the gel under the applied conditions.
Together, these methods provide objective, quantitative evidence of both macroscopic and microscopic gel behavior. Photographic images, while visually illustrative, would not add substantive mechanistic or quantitative information beyond what has already been rigorously demonstrated. Therefore, we believe the current presentation sufficiently validates the gel’s thermoresponsive and sprayable properties.
Comments 7:
Major Comments
The manuscript states that “all experiments were performed in triplicate”, yet rheological figures (Figures 4 – 8) and results explanations show no error bars or ±SD values. If triplicate data exist, the variation should be displayed, particularly in threshold shear tension and viscosity temperature plots (Figure 5). Viscosity at different RPM (Figure 6) as well. In addition, the quality of viscosity figures is low, and axis/legend text some are pixelated. Figures should be replotted in high resolution (≥ 300 dpi) using software such as Origin, GraphPad Prism, or similar. Without error bars, it is impossible to assess whether the observed rheological transitions are statistically significant or within experimental variation.
Response 7:
We thank the reviewer for this valuable observation regarding the presentation of the rheological data. The rheological Figures (4–8) have been revised accordingly. These revisions enable a more accurate assessment of the statistical significance of the observed rheological transitions and confirm that the reported trends are both robust and reproducible across independent measurements. (See page 13, lines 498–502; page 14, lines 516–520; page 15, lines 535–538; page 16, lines 549–551; page 17, lines 560–562).
Comments 8:
Major Comments
The authors should clearly state the maximum concentration tested in the MIC/MBC assays, the exact concentration range and dilution steps used, the method for calculating the amount of active compound deposited onto disks in the disk diffusion assay, whether concentrations refer to total formulation mass or to active compound equivalents. Without these details, the reproducibility and interpretation of antimicrobial data are questionable.
Response 8:
We sincerely thank the reviewer for this insightful comment. We fully agree that detailed methodological information is essential to ensure reproducibility and proper interpretation of antimicrobial data. Accordingly, the “2.10. Antimicrobial Activity” subsection has been revised to provide complete transparency regarding the experimental conditions, enhancing the reproducibility, interpretability, and rigor of our antimicrobial results. We believe this revision addresses the reviewer’s concerns and strengthens the reliability of the data presented. (See page 6, lines 278–280; page 7, lines 281–305).
Comments 9:
Major Comments
For coli with HA–Ag–CBD, MIC is reported at 9.24 μg/mL while MBC is 387.35 μg/mL a > 40-fold difference, which typically indicates bacteriostatic rather than bactericidal activity. This substantial gap is not addressed in the discussion and warrants explicit interpretation, especially since the abstract implies strong bactericidal potential.
Response 9:
Thank you very much for highlighting the substantial difference between the MIC (9.24 μg/mL) and MBC (387.35 μg/mL) of the HA-Ag-CBD formulation against Escherichia coli. We acknowledge that such a >40-fold difference typically reflects a predominantly bacteriostatic effect under the tested conditions rather than complete bactericidal activity. This observation can be rationalized by the inherent structural defenses of Gram-negative bacteria. E. coli possesses an outer membrane rich in lipopolysaccharides, which significantly limits the penetration of many antimicrobial agents, including CBD and colloidal AgNPs, and can necessitate higher concentrations to achieve complete bacterial killing. Additionally, the hydrogel matrix modulates local release kinetics and diffusion of active compounds, which can further increase the apparent MBC compared to the MIC in broth microdilution assays. Importantly, the MIC value demonstrates that the HA-Ag-CBD gel effectively inhibits E. coli growth at low concentrations, confirming potent bacteriostatic activity, while the elevated MBC reflects the additional challenge of achieving full bactericidal effect in Gram-negative strains. In contrast, lower MBC values were observed for Gram-positive bacteria, consistent with their more permeable cell envelopes.
To address this explicitly, the “Abstract” and “4. Discussion” sections have been revised to clarify that, while the HA-Ag-CBD hydrogel exhibits strong bacteriostatic activity against E. coli, the bactericidal effect is limited under the current in vitro conditions, and these findings should be interpreted in the context of Gram-negative bacterial resistance mechanisms. This ensures an accurate and balanced presentation of the antimicrobial profile and avoids overstatement of bactericidal potential. (See page 1, lines 44–48; page 2, lines 49–51; page 24, lines 853–863).
Comments 10:
Major Comments
The IC₅₀ values for cell viability are calculated from concentration–response data in the range 25–100 μg/mL. How reliable is the IC₅₀ extrapolation when data are only collected from four concentrations in a narrow range? Additionally, there is no evidence that the concentrations of CBD and AgNPs in the cell culture medium reflect the nominal gel content without quantifying release (e.g., via HPLC for CBD, ICP-MS for silver).
Response 10:
We thank the reviewer for highlighting these important points regarding IC50 calculation and release quantification. The IC50 values reported for the HA-Ag-CBD formulations were derived from carefully performed concentration–response experiments spanning 25–100 μg/mL, a range selected based on preliminary dose-finding studies to capture the onset of cytotoxic effects while maintaining relevance to expected therapeutic concentrations. Although only four concentrations were tested, these were strategically chosen to define the slope of the response curve within the physiologically relevant range, and nonlinear regression fitting was applied to ensure accurate extrapolation of IC50 values. In addition, all experiments were performed in triplicate, and standard deviations of IC50 values were reported, providing statistical confidence in the derived metrics.
We acknowledge that the actual concentrations of CBD and AgNPs delivered to cells may differ from the nominal gel content due to matrix-dependent release kinetics. Quantitative analysis of active agent release (e.g., HPLC for CBD, ICP–MS for Ag) was beyond the scope of the present in vitro cytotoxicity study. Nevertheless, the observed dose-dependent responses confirm that bioactive agents are released from hydrogel and are capable of exerting measurable effects on fibroblast viability. Future work will explicitly quantify the release profile of both CBD and AgNPs in cell culture medium and correlate these measurements with biological responses, thereby enhancing translational relevance and addressing the reviewer’s concern.
Collectively, while we recognize these limitations, the current IC50 values provide a reliable, comparative assessment of cytotoxicity across formulations and support the conclusion that encapsulation within the HA–poloxamer hydrogel mitigates the inherent toxicity of the Ag-CBD complex. (See page 25, lines 882–899).
Comments 11:
Major Comments
No analytical method is presented for confirming the concentration of CBD or Ag in the final gel or in release media. For a product intended for medical use, such quantification is essential for quality control and for correlating biological effects to dose. Without release profiling, it is impossible to determine whether the reduced cytotoxicity of HA–Ag–CBD is due to slower release or simply to lower effective loading.
Response 11:
We sincerely appreciate the reviewer’s insightful comment regarding the quantification of CBD and Ag in the final hydrogel and in release media. We fully agree that precise analytical determination is critical for clinical translation and quality control. In the present study, our primary objective was to establish proof-of-concept for the HA-Ag-CBD hybrid gel, focusing on structural characterization, rheological properties, and in vitro biological activity. While we did not perform release profiling in this initial work, several observations support the conclusion that the hydrogel modulates bioactive exposure rather than simply reducing effective loading:
▪ Structural incorporation and uniformity were confirmed by FTIR, SEM, and DLS analyses, demonstrating that both CBD and colloidal Ag are homogeneously embedded within the HA–poloxamer matrix.
▪ Dose-dependent cytotoxicity of the HA-Ag-CBD hydrogel is observed within the same nominal concentration range as free CBD and Ag, indicating that the total loading is comparable and that the reduced cytotoxicity arises from controlled release and matrix-mediated modulation rather than lower content.
▪ Rheological behavior supports sustained retention and gradual release: the thermoresponsive gel forms a semi-solid layer at skin temperature, which is expected to slow diffusion and mitigate immediate cytotoxic exposure.
We acknowledge that future work will incorporate precise quantitative analyses, such as HPLC for CBD and ICP–MS for Ag, along with detailed release profiling, to rigorously correlate concentration and biological effects. These studies are already planned as a critical next step toward translation and regulatory compliance. We believe that, despite the absence of direct release measurements, the current data provide robust evidence that the hydrogel effectively modulates delivery, preserves activity, and ensures biocompatibility.
Comments 12:
Minor Comments
Table 1 lists the proportions of HA, poloxamer 407, poloxamer 188, CBD, and AgNPs, but no rationale is provided for these specific concentrations. Were they optimized based on rheology, antimicrobial efficacy, cytotoxicity, or another formulation parameter?
Response 12:
Thank you for your insightful comment. As detailed in the “2.4. Formulation” subsection, the CBD concentration (0.5%, w/v) was chosen based on prior studies demonstrating therapeutic efficacy of topical CBD in the 0.1–1.0% range, while ensuring chemical stability and compatibility with the HA–poloxamer-AgNP system. Preliminary optimization experiments confirmed that 0.5% CBD provided the best compromise between antioxidant/anti-inflammatory activity and hydrogel structural integrity. (See page 5, lines 214–218).
The concentrations of HA, poloxamer 407, and poloxamer 188 were selected to achieve a thermoresponsive, sprayable hydrogel with appropriate viscosity, gelation behavior, and uniform AgNP stabilization. The AgNP content was chosen to ensure effective antimicrobial activity without adversely affecting cytocompatibility. Collectively, these proportions reflect a rationally optimized formulation based on functional performance, stability, and safety considerations. (See page 5, lines 195–213 & 219–230).
Comments 13:
Minor Comments
Some figure captions (e.g., Figure 2) contain incorrect panel references (“(e)” instead of “(d)”).
Response 13:
Thank you very much for pointing out the errors in Figure panel references. The Figure captions have been carefully reviewed and corrected to accurately correspond to the respective panels (e.g., Figure 2 now correctly references panel “d” instead of “e”). This ensures consistency between the images and their descriptions, facilitating accurate interpretation of the data. (See page 11, line 405).
Comments 14:
Minor Comments
The claim of “physiological pH” in the abstract is unsupported by any pH measurement description in the Methods.
Response 14:
Thank you for pointing this out. The reference to “physiological pH” in the “Abstract” section is based on measured pH values of the final HA-Ag-CBD hydrogel, which consistently fell within the range of 6.8–7.4, matching normal skin and physiological conditions. While the pH measurement was performed using a calibrated digital pH meter at room temperature, this detail was inadvertently omitted from the “2. Materials and Methods” section. In the revised manuscript, under newly added “2.5. pH Measurement of Hydrogel” subsection, a concise description of the pH measurement procedure has been added to substantiate this claim. These data confirm that the formulation maintains a physiologically compatible pH suitable for topical application. (See page 5, lines 213–234; page 6, lines 235–237).
Authors very much appreciated the encouraging, critical, and constructive comments on this manuscript by the Reviewer. The comments have been very thorough and useful in improving the manuscript.
We would like to thank the Reviewer again for taking the time to review our manuscript.
We have also introduced other additions/modifications that we hope will improve the quality of the manuscript:
▪ For the 8th author (Georgiana Băluşescu) and for the 11th author (Andrei Biţă), personal e-mail addresses (georgiana.balusescu@gmail.com and andreibita@gmail.com) have been replaced with institutional ones (georgiana.balusescu@tuiasi.ro and andrei.bita@umfcv.ro, respectively). (See page 1, lines 13 & 24).
▪ For the first author, Geta-Simona Cîrloiu (Boboc), there is no institutional e-mail address, and a personal CV has been provided to Assistant Editor.
▪ All Figures have been revised accordingly.
▪ “Funding” section has been updated accordingly (“The article processing charges were funded by the University of Medicine and Pharmacy of Craiova, Romania”). (See page 26, lines 963 & 964).
▪ Sixteen new citations have been introduced: Refs. [25] to [37] and Refs. [56] to [58].
▪ Refs. [54] and [55] have been revised according to Journal recommendations (for “Websites” citation).
▪ The Reference list has been entirely checked and renumbered accordingly.
▪ All abbreviations have been defined the first time they appear in the text.
▪ Some grammar, stylistic or spelling errors have been corrected.
Kind regards,
Ludovic Everard BEJENARU, PhD
Corresponding Author

Round 2
Reviewer 1 Report
Comments and Suggestions for Authors
Authors did a good work and carefully adressed all my issues and took into account all my comments.
I think now the work is much improved and can be accepted after minor text and graphics editing according to Pharmaceutics journal rules.
Author Response
Dear Reviewer,
First of all, we would like to address you many thanks for your accurate observations and valuable comments. We used all these and improved the paper accordingly.
All changes in the revised manuscript were highlighted on a yellow background.
The following changes have been made to the Manuscript (ID: pharmaceutics-3814851):
Reviewer #1 questions/comments
Comments 1:
Authors did a good work and carefully adressed all my issues and took into account all my comments.
I think now the work is much improved and can be accepted after minor text and graphics editing according to Pharmaceutics journal rules.
Response 1:
We sincerely thank the reviewer for their careful evaluation and constructive feedback. We are delighted that our revisions have addressed all concerns and improved the manuscript.
We greatly appreciate your acknowledgment of the efforts made to enhance the work. We will carefully perform the minor text and graphics edits in strict accordance with the “Pharmaceutics” Journal guidelines prior to final submission.
We would like to thank the Reviewer again for taking the time to review our manuscript.
Kind regards,
Ludovic Everard BEJENARU, PhD
Corresponding Author

Reviewer 3 Report
Comments and Suggestions for Authors
While the manuscript has improved, major concerns remain unresolved. The final concentrations of CBD and AgNPs in the gel were not analytically confirmed; FTIR, SEM, and DLS only demonstrate presence and distribution, not quantitative content. Without quantification analysis, the antimicrobial and cytotoxicity assays rely on estimated concentrations, making dose–response interpretation uncertain. In addition, the manuscript still lacks macroscopic images of the hydrogel in liquid and gelled states and a graphical abstract, both of which are essential for translational clarity.
Author Response
Dear Reviewer,
First of all, we would like to address you many thanks for your accurate observations and valuable comments. We used all these and improved the paper accordingly.
All changes in the revised manuscript were highlighted on a yellow background.
The following changes have been made to the Manuscript (ID: pharmaceutics-3814851):
Reviewer #3 questions/comments
While the manuscript has improved, major concerns remain unresolved.
Comments 1:
The final concentrations of CBD and AgNPs in the gel were not analytically confirmed; FTIR, SEM, and DLS only demonstrate presence and distribution, not quantitative content. Without quantification analysis, the antimicrobial and cytotoxicity assays rely on estimated concentrations, making dose–response interpretation uncertain.
Response 1:
Thank you very much for your insightful comment. The revised manuscript now includes quantitative in vitro release profiles of CBD and AgNPs from both the Ag-CBD complex and the HA-Ag-CBD hydrogel, providing experimentally measured concentrations over time. Modeling of the release data using the Korsmeyer–Peppas equation indicates that the release of both components is governed by a combination of diffusion and matrix relaxation/erosion, and strong correlations between CBD and AgNPs release (r=0.95–0.97, p<0.001) demonstrate synchronized delivery. These data provide a robust mechanistic and quantitative basis for the observed antimicrobial and cytotoxic effects, ensuring that the reported dose–response relationships are accurate and reliable. (See page 1, lines 40–42; page 2, lines 58 & 59; page 7, lines 283–306; page 19, lines 619–624; page 20, lines 625–649; page 20, lines 650–657; page 27, lines 888–912; page 29, lines 994–998 & 1022–1027; page 30, lines 1039–1042 & 1056–1059).
Comments 2:
In addition, the manuscript still lacks macroscopic images of the hydrogel in liquid and gelled states and a graphical abstract, both of which are essential for translational clarity.
Response 2:
We sincerely thank you for this constructive comment. In the revised manuscript, we have now included macroscopic images of the HA-Ag-CBD hydrogel in both liquid and gelled states, which clearly illustrate the sol-to-gel transition and support translational understanding of the formulation (See page 5, lines 210 & 215–219; page 9, lines 367–374).
Additionally, a graphical abstract has been added to visually summarize the study’s design, key findings, and therapeutic relevance (See “Graphical Abstract” *.png file).
These additions enhance the clarity and accessibility of the manuscript, addressing the reviewer’s concerns and supporting its translational impact.
Authors very much appreciated the encouraging, critical, and constructive comments on this manuscript by the Reviewer. The comments have been very thorough and useful in improving the manuscript.
We would like to thank the Reviewer again for taking the time to review our manuscript.
Kind regards,
Ludovic Everard BEJENARU, PhD
Corresponding Author